# The Effect of Using Augmented Reality Technology in Takeaway Food Packaging to Improve Young Consumers' Negative Evaluations

Chao Gu [1], Tingting Huang [2], Wei Wei [3], Chun Yang [4], Jiangjie Chen [4], Wei Miao [3], Shuyuan Lin [5], Hanchu Sun [6] and Jie Sun [7],*

1   Department of Culture and Arts Management, Honam University, Gwangju 62399, Republic of Korea
2   School of Art and Design, Minnan Science and Technology University, Quanzhou 362300, China
3   School of Textile Garment and Design, Changshu Institute of Technology, Changshu 215500, China
4   School of Design, Jiangnan University, Wuxi 214122, China
5   Department of Media Design, Tatung University, Taipei 104, Taiwan
6   The Faculty of Industrial Design Engineering, Delft University of Technology, 2628 CE Delft, The Netherlands
7   College of Arts and Design, Zhejiang A&F University, Hangzhou 311300, China
*   Correspondence: sunjie@zafu.edu.cn

**Abstract:** This paper examines the use of augmented reality technology in the design of packaging for takeaway food to assist in marketing. The research is divided into three studies for progressive investigation and analysis. Study 1 collected 375,859 negative evaluations of food delivery from the Internet and explored the main reasons that may have impacted the user's evaluation by Latent Dirichlet Allocation topic modeling. Study 2 evaluated the effectiveness of augmented reality packaging by surveying 165 subjects and comparing it with traditional packaging. We conducted a survey of 1603 subjects in Study 3 and used the technology incentive model (TIM) to analyze how augmented reality technology positively impacts food delivery marketing. It has been established that packaging will influence the negative perception of consumers about buying and eating takeout food. Specifically, augmented reality technology can improve negative evaluations by providing a more conducive user experience than traditional packaging. According to our findings, augmented reality technology has improved the consumers' perception of interaction, perceived vividness, and novelty experience, and achieved the aim of promoting takeaway food retail by improving negative evaluations posted by users.

**Keywords:** augmented reality; takeaway food; user evaluation; technology incentive model; marketing

## 1. Introduction

### 1.1. Research Background

Increasingly, catering services are being delivered through the Internet as technology advances. As an example, in 2020, China will have more than 456 million users of online food ordering, and the total turnover of food orders via the Internet will reach CNY 811.94 billion [1]. Due to this opportunity, many catering operators are gradually considering food delivery as a major source of income. Even high-end brands, such as Michelin-starred restaurants and star hotel groups, have tried to introduce takeaway services in Beijing and Shanghai [2]. In Taiwan, Ministry of Digital Affairs has been set up with a budget of over New Taiwan dollar 20 billion for 2022. It has immediately considered developing a food delivery platform and promoting consumption of food delivery as one of its main responsibilities since its establishment. The COVID-19 global pandemic may increase the risk of infection for consumers dining in stores during this period. Food delivery, on the other hand, can provide non-contact delivery services and a variety of dining locations, which will help meet the needs of customers for epidemic prevention as well as improve the performance of catering companies [3].

Previous studies have discussed ways to increase consumers' willingness to purchase food delivery. According to Shi et al. [4], the primary way to develop food delivery economies is to increase population density around the business circle, the number of caterers, and transportation convenience. Although the food delivery service has a particular radiation radius, the turnover is still impacted by the location of the retailer. Location characteristics lead to a constant number of customers in the area with stable demand. In the event that consumers have a negative evaluation of food delivery, how to change this perception and obtain the opportunity to sell again becomes increasingly important. The provision of takeaway service may result in losses in the event of a small order quantity [5]. Since COVID-19 has spread continuously, consumers have become more cautious in their purchasing decisions [6]. A lower number of negative evaluations means more orders, which demonstrates the connection between consumer comments and restaurant operators' survival.

Chinese young consumers are among the first to turn takeaway food into a daily habit [7]. A survey of 1000 college students in Nanjing conducted in 2019 revealed that at least 71.45% have used food delivery services for at least two years, and 85.1% have used them more than once a week [8]. Therefore, the food delivery industry needs to pay particular attention to young people's consumption choices. In addition, takeaway food sales may be affected by packaging [9]. It is believed that packaging design often influences the willingness of consumers to purchase a product [10]. Through packaging, takeaway merchants can communicate food information, brand concepts, and other relevant information to consumers. In particular, discussions on health, environmental protection, and other issues have gradually begun to affect the behavior and decision-making of some consumers when it comes to food packaging [11].

Several catering companies have recognized the importance of improving consumer evaluations, and several designers have proposed design suggestions for takeaway packaging. According to Spence and Velasco [12], packaging color conveys sensory information to consumers, such as taste, as well as more abstract images of brands. Simmonds et al. [13] found that transparent packaging allows consumers to see the food and also increases their willingness to purchase. In designing takeaway packaging, restaurants and designers often pay close attention to color, form, and function. However, as far as we know, there have been very few attempts to incorporate augmented reality technology into takeaway packaging. As augmented reality has evolved in recent years, it has gradually become a mature technology. A number of studies have demonstrated its positive impact on marketing. As an example, the virtual shoe test function developed using augmented reality has significantly increased the likelihood of consumers purchasing footwear [14]. It has been found that consumers who have experienced augmented reality have a more positive perception of beauty products they purchase [15]. Accordingly, it may make sense to design takeaway packaging using augmented reality technology as part of a marketing strategy.

Many marketing campaigns have already demonstrated the potential of augmented reality technology. However, there is currently a lack of research systematically evaluating the significance and value of incorporating augmented reality into takeaway packaging design. Considering that restaurant operations rely on a fixed customer base within a certain range, we are especially concerned about young consumers who have already expressed negative views regarding food delivery service. In this study, we conduct an extensive search of negative evaluations from food delivery platforms and summarize the most likely causes of negative perceptions. In addition, the study makes reference to the technology incentive model used in previous literature to evaluate the interactive experience of augmented reality technology [16]. We examine the mechanism and effect of improving negative evaluations by using augmented reality packaging by introducing a quantitative theoretical model. This research extends theoretical models in the field of augmented reality. It is equally urgent and necessary from a practical management perspective as it assists restaurants and designers in gaining a better understanding of the impact of augmented reality packaging on consumers. The majority of existing food

marketing research focuses on the operation and management of restaurants [17], the impact of takeout on the environment [18], and food safety issues [19,20]. The findings of our study complement the lack of literature relating to the theoretical development of interaction effects in the design of food packaging. The use of augmented reality has become increasingly popular in recent years as a highly strategic and interdisciplinary marketing tool [21]. This suggests that augmented reality technology has potential to be used in marketing efforts to appeal to consumers and improve marketing effectiveness. Our study provides suggestions and references for the future effective use of this interactive medium in the field of food marketing.

*1.2. Research Purposes*

To assist restaurants in improving the performance of food delivery services in retail, it is imperative to improve the negative ratings of consumers in the delivery region. As augmented reality technology has been proven beneficial to marketing in numerous industries and activities, this study examines whether application of this technology in takeaway food packaging can improve consumer negative evaluations, as well as how consumer behavior is affected. The researchers conducted three studies to evaluate whether the application of augmented reality technology to takeout packaging would help improve consumers' negative perceptions of these products. Figure 1 illustrates the research process. As part of Study 1, we conducted topic modeling using Latent Dirichlet Allocation to summarize the main causes of negative evaluations. We conducted Study 2 to explore whether augmented reality packaging could prove helpful for consumers by selecting a random product packaging sample and creating augmented reality samples to compare the perceived difference between ordinary packaging and augmented reality packaging. In Study 3, we developed a structural equation model based on a technology incentive model in order to theoretically verify the effect of augmented reality packaging on negative evaluations.

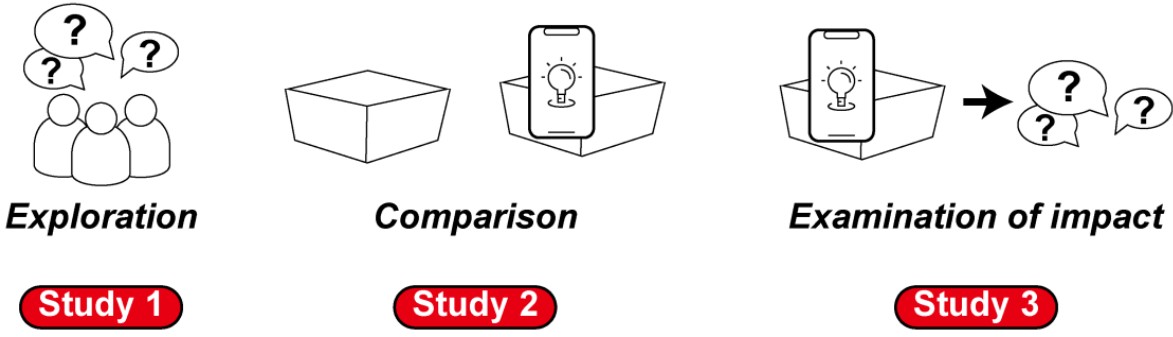

**Figure 1.** Research process.

## 2. Literature Review and Research Hypotheses

In this study, a quantitative model is developed to assess the effect of augmented reality packaging on improving negative evaluations. The study examines the perception and behavior of consumers when augmented reality packaging is used. The study examines the technology incentive model, satisfaction, and purchase intention. A relational framework is proposed to describe path relationships among constructs, and all hypotheses are tested. Figure 2 shows the hypothesized model. According to the extended unified theory of acceptance and use of technology (UTAUT2), gender, age, and experience of consumers may be important moderating factors [22]. There has been a huge impact of this theoretical model in the field of user research [23]. Thus, we examined the moderating effects of these three variables for each path relationship in our study.

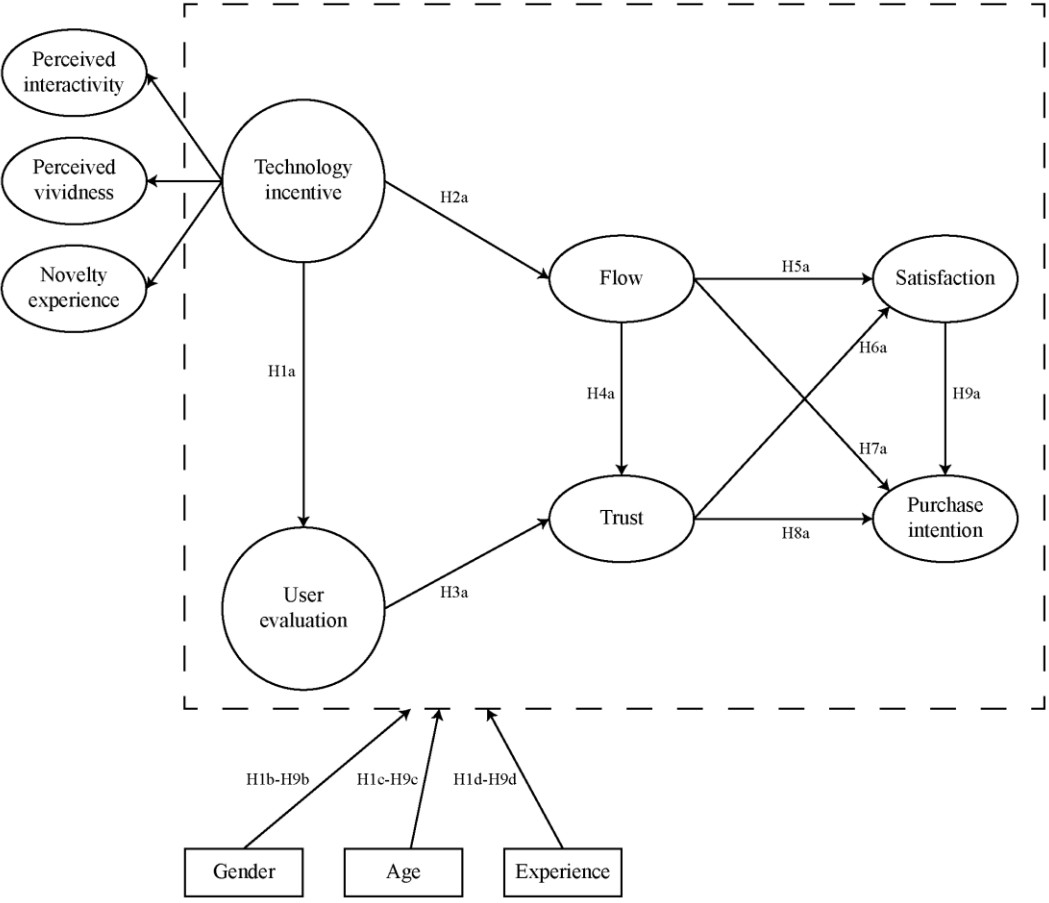

**Figure 2.** Research hypotheses.

### 2.1. Technology Incentive Model

The technology incentive model was first proposed as a means of verifying the perception and preference of users when using drones for matte painting in first-person perspective [16]. The use of first-person view drones provides an interactive experience environment as a new technology. The interaction between users motivates them and influences their behavior [24]. There are many areas in which technology can provide valuable assistance, due to the new experiences it provides. As an example, interactive education may enhance students' interest in learning [25]. The efficiency of military operations may be improved by enhancing the interaction between man and machine [26]. It can also provide a more immersive experience for museum visitors [27]. Some studies have also confirmed that the adoption of new technology in the marketing field is closely related to the positive perception of the consumers towards it. In Lithuania, by way of example, e-marketing tools contribute significantly to enhancing the management of travel communication and promoting the intention of Lithuanian consumers to travel [28]. The luxury retail industry can benefit from the use of new media for marketing communications [29]. As a result of these studies, it is evident that technology has tremendous potential for development and appropriate application in the marketing domain. New technologies may facilitate consumer decisions based on the premise of positive perceptions, realizing the importance of technology in marketing [30]. To verify the relevance of technical incentives in the practical application of augmented reality packaging, we include them as an important antecedent variable in the model.

#### 2.1.1. Perceived Interactivity (PI)

A user's perceived interactivity is defined as their ability to modify the content of a human–computer interaction environment [31]. Thus, permissions granted by the system

and developers are one of the factors influencing perceived interaction. Lu et al. [32] propose that users jointly evaluate perceived interactivity on the basis of three dimensions: control, responsiveness, and communication. Thus, the purpose of this document is to describe what users can do in the system, how they experience the process, and what changes are resulting from the results. Yang et al. [33] suggest that the real-time interactive experience in the online environment influences the perception of interaction by the user. Furthermore, in a study of website design, perceived interactivity was viewed as an indicator of how much the users were able to influence the website's design [34]. According to this study, perceived interactivity refers to the user's experience of participation and co-creation provided by augmented reality packaging. Perceived interactivity can be considered as an empirical evaluation because it merges process, characteristics, and perception [35].

2.1.2. Perceived Vividness (PV)

The term perceived vividness refers to the clarity with which users convert and link to relevant images in response to stimuli [36]. A higher level of vividness enhances the sense of realism experienced by users [37]. As technology evolves, new communication channels are being established between businesses and consumers. As digital technology develops, consumers are able to experience the media environment virtually and evaluate its vividness [38]. Similarity between the virtual environment and the real environment may contribute to the formation of consumers' perception of vividness. From the perspective of design, perceived vividness is closely related to product presentation, which may influence consumer behavior and decision-making [39]. We used perceived vividness in this study to assess the authenticity of augmented reality packaging when it conveys marketing messages to users both virtually and physically [40].

2.1.3. Novelty Experience (NE)

Technological advancements, disruptive business models, or creating a new image are all means of inducing novelty in consumers [41]. It is a feeling that is closely related to the previous experience. The experience of new things and events that are different from those in everyday life has been shown to contribute to positive perceptions in a variety of studies [42]. A consumer's perception of technological novelty will also influence how he or she views technology to a certain extent, according to the theory of diffusion of innovation [43]. The impact of technological advancement is not always positive. When a new technology does not appear to have superior functions or is difficult to use, consumers may be inclined to reject it [44]. An assessment of the novelty and originality of a product is determined by comparing the experiences of the user with the experiences of other users [45]. This study compares consumer experiences of augmented reality packaging with past experiences using novelty experience, including new, unique, and different experiences consumers have as a result of technological advances [46].

Second-order constructs of the technology incentive model include perceived interactivity, perceived vividness, and novelty experience [16]. The research model developed in this study was based on this theoretical framework. The evaluation of interaction design by users is mainly determined by these three dimensions, which are considered to be important new attributes within the interaction process [46]. Based on the literature, we further hypothesize:

**Hypothesis 1 (H1):** *Technology incentive has a positive impact on user evaluation (H1a); there is a moderating effect of (H1b) gender, (H1c) age, (H1d) experience on the path of technology incentive and user evaluation when takeaway food uses augmented reality to design the packaging.*

### 2.1.4. Flow (FL)

A flow experience occurs when people pay full attention to their actions [47]. Flow is very similar to immersion in that it is characterized by focus and positive feelings. The most important difference is that immersion is usually hierarchical, and users at different levels of immersion will experience different feelings and characteristics. In contrast, flow experience refers to a state of complete absence or complete absence of information [48]. As soon as the flow is initiated, the user's positive perception is extreme and complete [49]. As a consequence, the flow requirement must also meet the main condition of the premise of interaction in order to fulfill the requirements of the flow. A balance must be achieved between the user's perceived ability and the difficulty of the challenge, along with clear goals and immediate feedback [50]. A flow experience occurs when a user interacts with devices or programs with input and output methods in real-time. When in a flow state, the user is characterized by reduction in self-awareness, reduced response to stimuli, and a sense of control over their environment [51]. Flow experience is used in this study to evaluate how consumers interact with augmented reality packaging. In previous studies, technical interactive behaviors have been demonstrated to generate positive perceptions, including states that trigger flow experiences in users [52]. A similar influence path may exist when consumers purchase takeaways packaged in augmented reality. The following hypotheses are proposed based on the literature reviewed in this study:

**Hypothesis 2 (H2):** *Technology incentive has a positive impact on flow (H2a); there is a moderating effect of (H2b) gender, (H2c) age, (H2d) experience on the path of technology incentive and flow when takeaway food uses augmented reality to design the packaging.*

### 2.1.5. Trust (TR)

When consumers believe that service providers are reliable, trustworthy, and capable of fulfilling their commitments, trust is formed [53]. In the context of food delivery, trust considerations include a wide range of factors, such as food safety. Consumers place greater emphasis on personal safety during the COVID-19 period when purchasing food, and are willing to pay more for methods that reduce risk [54]. However, consumers do not care if the food is delivered from a merchant with a physical store, and the perception of food delivery platforms does not influence purchase decisions [55]. In this regard, it appears that the facility conditions of restaurant outlets do not appear to affect consumer trust. Based on the results of the trust assessment, we can understand how consumers perceive businesses' integrity and honesty when purchasing food [56]. Restaurants may be able to enhance trust by portraying an image of integrity in addition to improving the quality of their dishes and services. A number of studies have indicated that trust is an important factor that may influence the behavior of consumers in a positive way [57]. Takeaway retail marketing should take into account the potential impact of trust. In this study, trust is defined as the degree to which consumers perceive takeaway restaurants to be reliable and honest in their food, service, and image. Research has shown that consumers' perceptions of food delivery can have a significant impact on restaurants' popularity and performance [58]. The flow experience of a product can have a positive impact on trust through the engagement of a customer brand [59]. In takeaway retail, the evaluation and flow of users may also contribute to trust. Several hypotheses have been proposed based on the literature.

**Hypothesis 3 (H3):** *User evaluation has a positive impact on trust (H3a); there is a moderating effect of (H3b) gender, (H3c) age, (H3d) experience on the path of user evaluation and trust when takeaway food uses augmented reality to design the packaging.*

**Hypothesis 4 (H4):** *Flow has a positive impact on trust (H4a); there is a moderating effect of (H4b) gender, (H4c) age, (H4d) experience on the path of flow and trust when takeaway food uses augmented reality to design the packaging.*

### 2.2. Satisfaction (SA)

As a concept, satisfaction is much discussed in the field of consumer behavior. In general, consumer satisfaction refers to a comparison between the expectations of a product or service and the actual experience [60]. There will be an expectation from the consumer for the entire dining process in advance. Once the actual experience is better than expected, it will result in higher satisfaction. Satisfaction is determined by whether the product or service provides a pleasure related to consumption feelings, including levels of under- or over-satisfaction [61]. Several researchers have investigated how to enhance consumer satisfaction in catering companies. For example, Chua et al. [62] conducted a systematic evaluation of restaurants from the perspective of user experience. They found that spectacular restaurants, creative menus, specialty restaurants, and gourmet creations created by world-renowned chefs will improve consumer satisfaction. We defined satisfaction in this study as consumers' overall satisfaction and pleasure with their takeaway food dining experience when using augmented reality packaging.

With the development of technology, new experiences brought by human–computer interaction are opening up new opportunities to enhance consumer satisfaction. According to Gu et al. [63], interactive narrative design effects and flow experience will have an adverse effect on the satisfaction of users. The flow experience provided by a restaurant's social networking site can be beneficial to brand marketing, as consumers will be more satisfied and have higher purchase intentions as a result [64]. Further studies have shown that trust has a positive impact on satisfaction. A survey on halal food found that consumers' trust was positively related to satisfaction [65]. Researchers found that consumers who rate food delivery reliability higher are more likely to be satisfied with the service [66]. Additionally, this study proposes the following hypotheses based on the literature:

**Hypothesis 5 (H5):** *Flow has a positive impact on satisfaction (H5a); there is a moderating effect of (H5b) gender, (H5c) age, (H5d) experience on the path of flow and satisfaction when takeaway food uses augmented reality to design the packaging.*

**Hypothesis 6 (H6):** *Trust has a positive impact on satisfaction (H6a); there is a moderating effect of (H6b) gender, (H6c) age, (H6d) experience on the path of trust and satisfaction when takeaway food uses augmented reality to design the packaging.*

### 2.3. Purchase Intention (PuI)

As a prediction of consumers' behavior, purchase intention refers to the likelihood that they will make purchases [67]. The willingness to buy is an indicator of how likely a consumer is to purchase goods or services, as the more inclined they are to buy goods or services, the higher their willingness to buy. In Castro et al. [68], purchase intention is defined as the probability that a consumer will make a purchase. This explains why consumer purchasing behavior is directly related to the results of revenue in the retail-oriented catering industry. Our study measures purchase intention as the likelihood that a consumer will order takeaway food after using an augmented reality takeaway package.

It has been noted that previous researchers have conducted a great deal of exploration and discussion in order to increase the likelihood of consumers buying food. Research on the modern technological environment suggests that flow experience has a positive impact on consumers' attitudes, ultimately increasing their likelihood of purchasing food [69]. It appears that a good experience with human–computer interaction can enhance the dining experience and promote consumption. Trust is also considered to be an important factor that affects purchase decisions positively [70]. It has been observed that certain companies pay special attention to how to gain consumers' trust in food retail, especially in the context of the COVID-19 pandemic. Another conclusion drawn from an analysis of the social media marketing characteristics of the fast food industry is that trust is closely related to purchasing decisions [71]. Since consumers have a higher level of trust, marketing purposes can be achieved more easily and brands can create a positive impression in the minds of

consumers [72]. All of these studies suggest that trust has an important impact on purchase intentions in our context. Additionally, we focus on the relationship between satisfaction and purchase intentions in our research. A study conducted by Konuk [73] concludes that the satisfaction of consumers with organic food affects purchase intentions. Based on the literature, the following hypotheses are proposed:

**Hypothesis 7 (H7):** *Flow has a positive impact on purchase intention (H7a); there is a moderating effect of (H7b) gender, (H7c) age, (H7d) experience on the path of flow and purchase intention when takeaway food uses augmented reality to design the packaging.*

**Hypothesis 8 (H8):** *Trust has a positive impact on purchase intention (H8a); there is a moderating effect of (H8b) gender, (H8c) age, (H8d) experience on the path of trust and purchase intention when takeaway food uses augmented reality to design the packaging.*

**Hypothesis 9 (H9):** *Satisfaction has a positive impact on purchase intention (H9a); there is a moderating effect of (H9b) gender, (H9c) age, (H9d) experience on the path of satisfaction and purchase intention when takeaway food uses augmented reality to design the packaging.*

*2.4. Consumer Behavior and Packaging Design*

Over the past few years, with the development of technology, a growing number of platforms have enabled consumers to post reviews online. Online reviews have gained increasing attention with the development of e-commerce, since they can reflect to some extent the potential purchase opportunities for consumers [74]. By reviewing consumer reviews, communicating with them, and making targeted changes in response, the manufacturer can gain a better understanding of consumers' opinions. Due to the peculiarities of online ordering, consumers cannot physically see the products they intend to purchase. It is possible for consumers who are unfamiliar with a product to effectively analyze and compare the expected results before actually making a purchase decision based on online product reviews [75]. Review sites can provide valuable information regarding a product's quality, and should be considered a reliable source of information. It is especially important when it comes to products that have no previous purchasing experience or cannot be easily accessed prior to purchase [76]. It is still important for consumers to read online reviews even when they are familiar with a product. Researchers have suggested that the continuous exchange of product, service, brand, and company information can be accomplished through the review data on the Internet among consumers who may have purchase behaviors, consumers who are purchasing, and consumers who have purchased products [77]. It can be concluded from this that consumer reviews are more than just an interaction between consumers, but also a means of communication between consumers and manufacturers. It is essential for manufacturers to be attentive to consumer comments in order to continue to make profits and improve the quality of products and services. Therefore, consumers and brands are constantly influencing one another [78]. Particularly when based on online reviews, manufacturers can quickly understand consumers' thoughts and respond efficiently. Thus, in marketing-related research, it is often concluded that consumers' ability to obtain online reviews and participate in reviews may contribute to higher product satisfaction [79]. It is imperative that manufacturers extract the key points of consumers' concerns from online reviews in a large number rigorously and effectively in order to execute a more effective marketing plan. Latent Dirichlet Allocation has emerged as one of the most widely used methods for topic discovery in recent consumer behavior research, especially for non-normalized data [80]. Our choice of this research method is primarily motivated by this consideration. A Latent Dirichlet Allocation topic model extracts topics from reviews using unsupervised learning methods. Among the advantages of this method is that it does not assume the grammatical attributes of the text while it is executing, and can be used effectively to identify themes in a large number of documents [81].

Once the negative comments of consumers have been understood, targeted adjustments need to be made. Furthermore, packaging design is one of the optional marketing tools. A packaging product's primary function is to protect, store, ship, sell, promote, provide service, and ensure security [82]. Especially for sales and promotional purposes, packaging design can assist manufacturers in communicating the attributes of their products to consumers in an effective and efficient manner. For instance, the color of the packaging can affect consumers' perceptions of the taste of the product [83]. In this regard, packaging design is an important method of conveying sensory characteristics to consumers through its visual elements. It is through these powerful and efficient means that consumers are able to effectively influence their consumption patterns [84]. Additionally, sound attributes in packaging may impact the consumer's experience and purchase decision [85]. In this regard, the auditory experience is also an important component of packaging design. Studies have shown that consumers perceive brands differently based on the complexity of packaging design. The simplicity or complexity of a design expresses different personalities of the brand and can easily influence consumers' perceptions of it [86]. An important role played by packaging design is to convey information to consumers and encourage them to purchase. Although price is certainly a non-negligible factor in influencing young people's purchasing behavior [87], design can also contribute in a sensory and cultural manner to marketing efforts in retail. Consequently, packaging design can have a significant impact on consumer behavior.

With the advancement of technology, a number of new interactive methods have shown promise in the field of marketing, such as augmented reality. It is necessary to conduct further research in the area of augmented reality as it relates to packaging images because the application effect still lacks theoretical support [88]. In the food industry, there have been relatively few attempts to apply augmented reality technology, while augmented reality is widely recognized as a technology that enriches consumer, food, and environmental interactions [89]. By scanning the packaging with their mobile phones, consumers are able to view and interact with virtual product models. It is possible to better understand the appearance, function, and use of a product through this feature [90]. Aside from that, some information can also be conveyed by augmented reality. For example, with augmented reality, consumers and packaging can interact more easily, thus improving the traceability of food production [91]. In addition, consumers will have a deeper and more comprehensive understanding of information as a result of this approach. A study found that consumers who use augmented reality packaging gain an increased understanding of food products compared to those who use static packaging [92]. Consequently, the application of augmented reality technology in packaging can contribute to restaurants' continuous success in food marketing. However, there is still a lack of theoretical research on consumer behavior to support this interactive marketing approach. This is the main purpose of our research.

## 3. Research Method

This research involves both qualitative and quantitative methods. This paper examines the effects of augmented reality technology applied to the packaging design of takeaway food in order to improve the perceived negative impact on young consumers. The detailed research methods for the three phases of the study are outlined below.

### 3.1. Finding Out What Drives Negative Consumer Evaluations of Takeout Food

In Study 1, from June 2021 to January 2022, we used web crawlers to collect negative comments on Chinese food delivery platforms. Crawled content consists of the city and province, the name of the store, the text content of the evaluation, and the store rating. It is necessary to remove non-Chinese characters that may be present in the text in order to facilitate the determination of the association probability between characters and achieve the purpose of text segmentation and topic modeling. After removing some comments that contain only non-Chinese characters, we split the complete text data into individual

words. The Jieba library is used as the basis for text segmentation. This is the mainstream Chinese text word segmentation lexicon. We utilize a number of current mainstream lexicons in order to remove meaningless characters from the message text in order to prevent interference from meaningless characters and achieve a more accurate analysis of the results.

The text data is analyzed using Python, with the gensim lda module estimating Latent Dirichlet Allocation models using our corpus of text data. Since the number of topics must be set manually, we introduce two parameters of perplexity and coherence to determine the optimal number of topics for the topic model. A smaller perplexity value indicates a better result, according to the derivation Formula (1) [93]. In addition, c value calculates the score using normalized pointwise mutual information (NPMI) and cosine similarity between words in the content vector. It can be concluded that the higher the value, the better [94].

$$Perplexity = \exp\left\{ -\frac{\sum_{d=1}^{M} \log p(w_d)}{\sum_{d=1}^{M} N_d} \right\} \tag{1}$$

Note: In the above listed Equation (1), $M$ represents the number of texts in the test corpus, $N_d$ represents the length of the number of words, and $p(w_d)$ represents the probability of the text.

The final step in our analysis is to exploit the Latent Dirichlet Allocation in order to extract topics. For topic analysis, this is a very effective method of unsupervised learning. A document is considered a collection of words, whose distribution is displayed as a probability distribution, and on the basis of the distribution, topic clustering is performed. By calculating the probability value, we are able to determine the lexical importance. The method for calculating topic frequency consists of multiplying the frequency of the vocabulary in the selected topic by the sum of all topics in the topic. The higher the probability value, the more important the word is to the topic, and it is therefore used as a reference for topic naming.

### 3.2. Design of Augmented Reality Packaging

In Study 2 and Study 3, Unity Vuforia was used to create augmented reality packaging. Currently, this is one of the most popular engines for designing augmented reality applications [95]. Augmented reality technology can be used to promote brands, introduce ingredients and cooking methods, introduce promotional information, and provide consumers with an interactive experience. There is a possibility that the development of augmented reality packaging will require higher costs. These costs will depend on factors such as prices and level of development in different countries, which requires further research. Upon equal distribution of the production costs, the unit price may not be significantly impacted by the production costs if there are a large number of sales. Hence, the purpose of this study is not to estimate the unit price of augmented reality packaging versus traditional packaging, but merely to assess the impact of this technology on consumer behavior. Based on the large number of negative evaluations on the food delivery platform, we selected three national chain catering brands that currently offer food delivery services. In general, these selected brands were receiving more negative evaluations than their peers, which is not just due to poor packaging design. In order to incorporate augmented reality technology into some of these brands' product packages, we designed augmented reality packages based on some of their product packages. For brand and copyright reasons, we present the augmented reality packaging renderings as hand-painted illustrations. The graphics are similar to the original images, but not identical, and are therefore for illustration only. Figure 3 shows the augmented reality package.

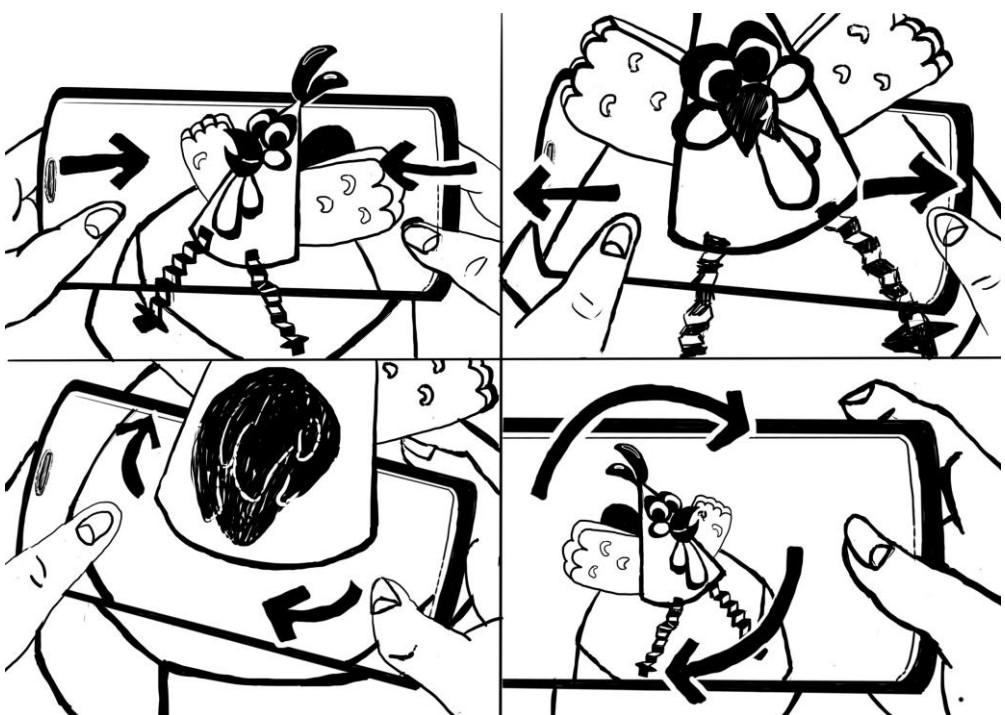

**Figure 3.** Sample of augmented reality packaging.

There are three main functions of augmented reality packaging. In the first part, consumers scan the pictures on the packaging using mobile phone software in order to present augmented reality to them, using pre-models and animations. In the second part, after scanning out the interaction effect, the model is controlled in size and rotation direction by gestures. Based on the premise that xSpeed and time are used as common coefficients, the program determines the angle of the rotation model by determining whether the touch position of the finger changes. To achieve two-finger slide control, the program grabs the two two-dimensional coordinate points after the two fingers first touch the screen, and the corresponding new two-dimensional coordinate point after the real-time movement, and determines whether the current gesture is zooming in or out by comparing the differences between the two coordinate points, and then multiplies the three-dimensional value of the model or adds the 1.025 constant. The third part consists of matching the sounds. We have made a pre-voice recording. If the target image is continuously tracked, the sound will be activated. If the target image is lost, the sound will be inactivated. By setting this parameter, when the consumer scans and switches different objects, the program plays the appropriate voice to create an interactive audio experience.

*3.3. Questionnaire Design*

Appendix A contains the questionnaire we used in Study 2 and Study 3. Of these, Study 2 investigated the three constructs of trust, satisfaction, and purchase intention, while Study 3 investigated all the constructs in the Appendix. We used items developed and validated in previous studies and designed a questionnaire with a five-point Likert scale. Five pre-test personnel were randomly chosen to read all questionnaires in accordance with the range of subjects tested in this study. As part of the evaluation process, they were asked to assess whether they fully understood the content of each item. After discussing these issues with the pre-test personnel, modifications were made to the Chinese expressions of the items in order to facilitate understanding by the participants.

*3.4. Collection of Data*

We conducted our survey in China. We mainly target young consumers since they order takeaway food more frequently and are more likely to be familiar with technology

products. As can be seen from the sampling results, marital status, income, education level, and occupation type are more representative of the general basic situation of Chinese youth. The basic information of the subjects is presented in Table 1. In Study 2, the subjects are recruited via the Internet to take part in the survey, with two measurements being repeated on each subject. In order to compare consumer trust, satisfaction, and purchase intentions for takeaway foods using original packaging and augmented reality packaging, we used multivariate analysis of variance (MANOVA). Initially, the subjects completed the questionnaire for the original packaging, then observed the augmented reality packaging and completed the second questionnaire. A total of 165 questionnaires were collected during the period of January to March 2022. As a check on the logic of the respondents, we set reverse questions, and judged their concentration based on the time they spent answering the questions. After excluding invalid samples, the remaining 104 valid questionnaires were used for data analysis. The effective rate of the questionnaire was 63.030%.

**Table 1.** Demographic characteristics of the respondents.

| Sample | Category | Number | Percentage (%) | Number | Percentage (%) |
|---|---|---|---|---|---|
| | | Study 2 | | Study 3 | |
| Gender | Male | 50 | 48.077% | 453 | 45.255% |
| | Female | 54 | 51.923% | 548 | 54.745% |
| Age | 20–29 | 57 | 54.808% | 577 | 57.642% |
| | 30–39 | 47 | 45.192% | 424 | 42.358% |
| Marriage status | Married | 62 | 59.615% | 651 | 65.035% |
| | Unmarried | 42 | 40.385% | 350 | 34.965% |
| Monthly Income | Below 4000 | 20 | 19.231% | 179 | 17.882% |
| | 4001–6000 | 22 | 21.154% | 200 | 19.980% |
| | 6001–12,000 | 42 | 40.385% | 425 | 42.458% |
| | 12,001–18,000 | 16 | 15.385% | 139 | 13.886% |
| | Above 18,001 | 4 | 3.846% | 58 | 5.794% |
| Education | Junior high school or below | 1 | 0.962% | 12 | 1.199% |
| | High school or secondary school | 3 | 2.885% | 33 | 3.297% |
| | Undergraduate or college | 88 | 84.615% | 864 | 86.314% |
| | Postgraduate or higher | 12 | 11.538% | 92 | 9.191% |
| Occupation | Civil servant | 6 | 5.769% | 85 | 8.492% |
| | Clerk | 37 | 35.577% | 462 | 46.154% |
| | Worker | 18 | 17.308% | 123 | 12.288% |
| | Public service agency | 14 | 13.462% | 94 | 9.391% |
| | Student | 12 | 11.538% | 135 | 13.487% |
| | Self-employed | 17 | 16.346% | 102 | 10.190% |

We also invite subjects to participate in the survey via the Internet in Study 3. In this study, we apply structural equation modeling to investigate the specific mechanism by which augmented reality can improve negative consumer ratings. There were 1603 questionnaires collected over a period of six months between June 2022 and September 2022. The subjects are required to complete the questionnaire after examining the augmented reality packaging samples. There are a total of 1001 valid questionnaires remaining after excluding the two groups of contradictory responses to reverse questions. The effective rate of these questionnaires is 62.445%.

## 4. Results

*4.1. Study 1—Finding Out What Drives Negative Consumer Evaluations of Takeout Food*

We collected 375,859 negative evaluation records of food delivery on the ELEME food delivery platform in order to understand the factors affecting consumers' negative judgments. In China, ELEME is one of the most popular platforms for food delivery [96]. The most common comment length is 249 characters, and the least common is 1 character. There are 2291 single-character comments. Among them, the common one-word comments are poor, slow, not good, bad, etc., which are still associated with negative evaluations of takeaways. The average length of a comment is 21.78 words. The most frequently appearing comments have a length of 3 characters, a total of 17,208, and a standard deviation of 22.88. Figure 4 illustrates the specific distribution of comments with the same word count. In total, 372,997 valid negative comments remain after removing the comments completely composed of non-Chinese characters.

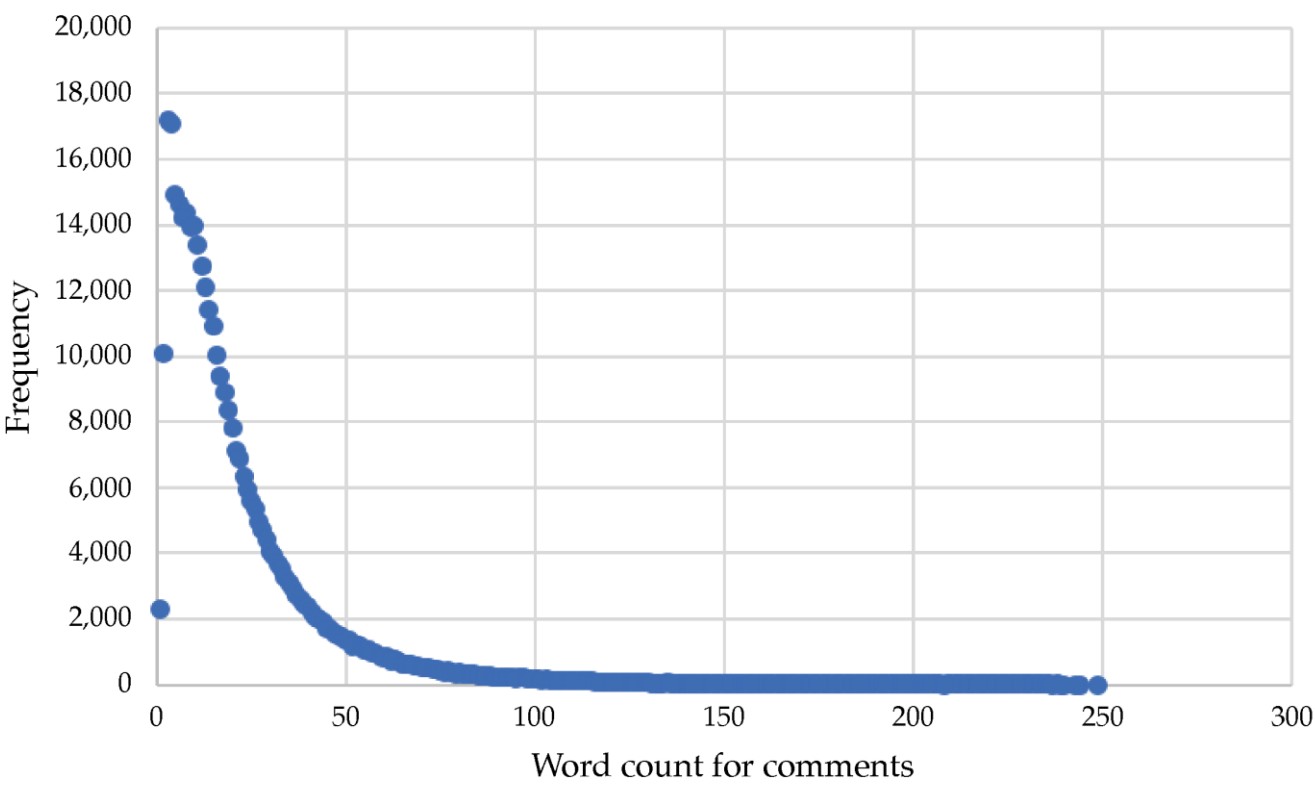

**Figure 4.** Commentary word count distribution.

In order to determine the optimal number of topics, we calculate the corresponding perplexity and consistency values, and Figure 5 shows the variation trend of the perplexity value with topic number. According to the results, the increasing trend of *p* value is not apparent when the number of topics is less than or equal to 10. In the case where the number of topics exceeds 10, the *p* value increases rapidly, indicating that a reasonable number of topics is less than 10.

Additionally, Figure 6 illustrates the change in consistency with the increase in the number of topics. When the number of topics exceeds 1, the value of c value begins to increase. There is very little increase in the value of c value when the number of topics increases from seven to eight, and from eleven to twelve. There are several rapidly rising phases located in topics 3, 6, 7, and 9. Therefore, the optimal number of topics was further determined to be probably these four cases.

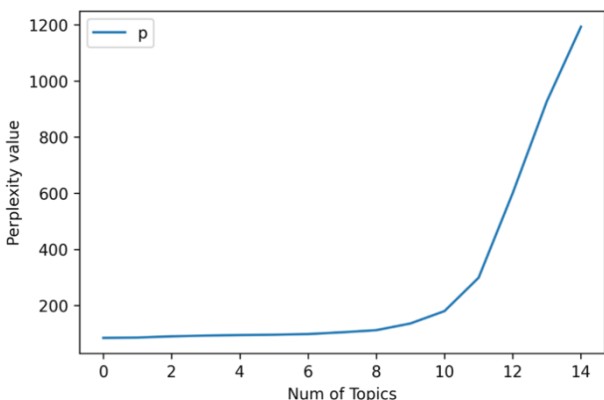

**Figure 5.** Perplexity trend.

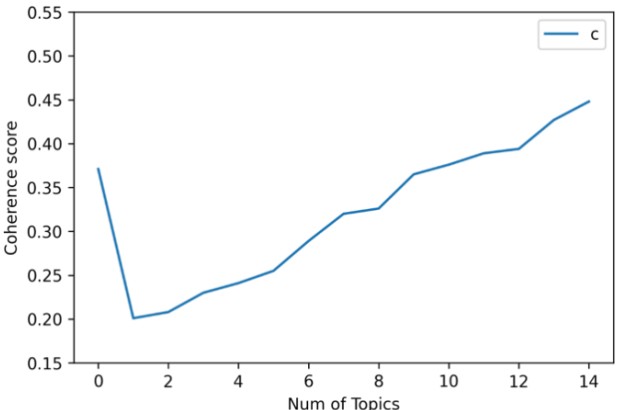

**Figure 6.** The trend of consistency.

We compared the overlap between topics in the case of 2 to 10 topics, as shown in Figure 7, to further determine the optimal number of topics. Interestingly, in topics with 4 themes of 3, 6, 7, and 9, there is no overlap between groups with only 3 topics. Accordingly, 3 topic numbers are used in the Latent Dirichlet Allocation topic modeling since there is a good distinction between topic concepts.

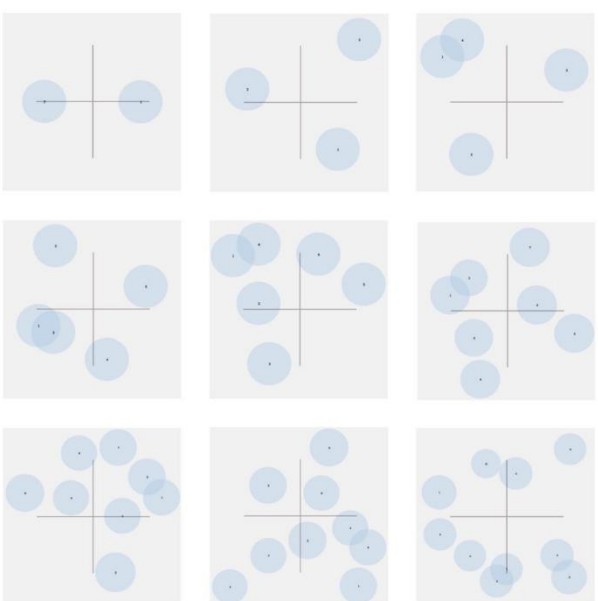

**Figure 7.** Cluster overlap from 2 to 10 clusters.

Table 2 shows the main constituent words of each topic. In the table, each topic is sorted by its probability value, which shows the top 10 words in terms of importance. It is noteworthy that some words appear in several topics at once, indicating that they contribute to each topic in an important manner. Considering that the same word appears in multiple topics in the previous studies using Latent Dirichlet Allocation topic modeling, this indicates that the topics may be influenced by one another [97]. In our opinion, this shows an excellent correlation between the topics and also provides some insight into the importance of these terms in marketing management. According to our analysis results, a total of 4 words are included in the top ten lists of multiple topics simultaneously, including taste, hair, unpalatable, and attitude. There was a simultaneous presence of flavors in all three subjects. Both themes 1 and 3 feature hair. The theme of unpalatable appears in both themes 2 and 3. Both themes 2 and 3 contain attitudes.

Using the semantics of the vocabulary within the topics, the probability value of the vocabulary, and the literature data, we identified three topics: quality of service, quality of food, and perceived value. The report indicates that when these three factors do not meet consumers' expectations when ordering and eating takeaway food, there is a high likelihood of negative feedback. This is also the basis of both Study 2 and Study 3; our research mainly focuses on whether adding augmented reality technology to packaging design will improve these negative comments.

It is significant that we have differentiated service quality and food quality in our research. Service quality and food quality are the two main components of restaurant operation [98]. Caterers have long valued the quality of their service and food. According to previous studies, food quality has a direct impact on consumer loyalty, and service quality has a positive impact on loyalty through perceived value and customer satisfaction [99]. These two concepts, however, are often confused as manifestations of consumers' perceived quality. Although perceived quality is defined differently in different studies, some researchers have begun to distinguish food quality and service quality as components of perceived quality in the area of catering, especially in the marketing research of food retail. Essentially, perceived quality refers to consumers' perceptions of food and service and can ultimately have a significant impact on their purchasing decisions [100]. Due to this, they are treated as two separate subjects in our study. The specific naming basis is as follows.

Topic 1. Quality of service (QoS).

It is important to note that the actual result of the service is composed of three dimensions, namely, the dimension of human interaction, the dimension of facilities, and the dimension of the content delivered [101]. We found that words are closely related to these three service-related dimensions in Topic 1. Takeaway food marketing emphasizes human interaction in the delivery time, online communication, and offline delivery experience. The consumer expects an ideal time and a positive human interaction experience when ordering food and receiving it. As a result, we identified hours, time, and experience as the most important factors in Topic 1. As the word with the highest probability value in Topic 1, chopsticks provide an explanation for the facility dimension. Furthermore, consumers' increasing emphasis on amenities is also reflected in the inclusion of tableware in the vocabulary. Consumers may not be concerned whether the merchant has a physical location when ordering food delivery [55]. A restaurant's traditional facilities, such as an inviting decor and comfortable seating, are no longer the first choice in terms of the facility dimension. Instead, consumers are more interested in the tableware that complements the meal. Hygiene appears in the vocabulary list. We consider it a component of both the facility dimension and the delivery content dimension, which is a quality attribute of the food itself. It is important to note that despite the fact that catering without a physical kitchen can reduce costs related to venue rental, decoration, and facilities and equipment, consumers continue to look for a trustworthy sanitation facility when choosing a catering service [8]. The last aspect of the delivery is the evaluation of the food itself, which includes words such as taste, hair, set meals, prices, etc.

**Table 2.** The results of Latent Dirichlet Allocation topic modeling.

| Topic 1 | | | | Topic 2 | | | | Topic 3 | | | |
|---|---|---|---|---|---|---|---|---|---|---|---|
| Word | Probability Value | Total Frequency | Topic Frequency | Word | Probability Value | Total Frequency | Topic Frequency | Word | Probability Value | Total Frequency | Topic Frequency |
| Chopsticks | 0.045 | 9177 | 6579.293 | Poor taste | 0.189 | 35,986 | 32,764.826 | Taste | 0.182 | 39,443 | 28,153.926 |
| Taste | 0.042 | 39,443 | 6072.040 | Flavor | 0.048 | 9255 | 8245.094 | Picture | 0.021 | 5953 | 3205.806 |
| Hair | 0.030 | 7532 | 4424.390 | Taste | 0.033 | 39,443 | 5722.826 | Diarrhea | 0.020 | 3745 | 3061.259 |
| Hygiene | 0.020 | 3642 | 2972.366 | Disgusting | 0.033 | 7145 | 5631.778 | Good value | 0.019 | 4055 | 2975.297 |
| Dinnerware | 0.020 | 3246 | 2891.386 | Attitude | 0.019 | 5563 | 3340.640 | Phone | 0.017 | 3385 | 2671.293 |
| Hour | 0.019 | 4437 | 2775.168 | Size | 0.017 | 5367 | 2883.718 | Cola | 0.017 | 4207 | 1975.477 |
| Set | 0.019 | 3912 | 1831.228 | Texture | 0.016 | 3537 | 2706.742 | Hair | 0.013 | 7532 | 1812.370 |
| Price | 0.015 | 3027 | 1786.107 | Driver | 0.015 | 4255 | 1696.774 | Box | 0.012 | 2631 | 1467.896 |
| Time | 0.013 | 3029 | 1565.857 | Bugs | 0.010 | 1906 | 1599.052 | Attitude | 0.012 | 5563 | 1234.145 |
| Experience | 0.013 | 2220 | 1320.659 | Clean | 0.009 | 1974 | 1523.079 | Poor taste | 0.010 | 35,986 | 1012.850 |
| Sum of topic frequency *: topic 1 = 146,161.200; topic 2 = 172,760.676; topic 3 = 154,657.150 | | | | | | | | | | | |

* The first 100 words in the topic were selected to calculate the sum of topic frequency, and the words after that were ignored because of their low frequency.

Topic 2. Quality of food (QoF).

Food quality refers to the degree to which food meets the needs of customers on an overall basis, which is considered one of the most important factors affecting customers' purchasing decisions [99]. Our findings in Topic 2 indicate that vocabulary is closely related to the definition of food quality and the important factors that influence food quality. The quality of food is determined by factors such as freshness, healthiness, and deliciousness [102,103]. According to the study, consumers place great emphasis on whether the food is fresh, healthy, and enjoyable. The following factors are associated with unpalatability, taste, nausea, weight, and mouthfeel. According to Zhang et al. [104], food safety is also an important part of food quality. Spoiled food can have an adverse effect on taste and even pose a risk of disease. In Topic 2, the words worm and clean reflect this concept. Besides the production process, the delivery process also plays a role in the variation in food quality assessments [105]. In our opinion, the delivery process can be divided into external factors and human factors for discussion. The delivery process may be adversely affected by external factors such as rough roads, congestion, vehicle accidents, and other events. A major factor affecting the human factor is the rider's attitude towards the takeaway, such as shaking it randomly, throwing it on the ground at random, or failing to notify consumers in time after delivery, leading to a lengthy processing time. Due to this, the words rider and delivery are considered part of the concept of food quality.

Topic 3. Perceived value (PVL).

Perceived value can be viewed as a multiple-dimensional aspect of the consumption process, focusing on the overall evaluation of the process [106]. There are six components of perceived value, including consumers' comparison of price and quality, self-satisfaction, aesthetic value, prestige value, transaction value, and hedonic value [107]. In addition to consumers' comparison results and understanding of products and prices, the definitions of the other concepts are based on emotions. Self-satisfaction may be defined as relaxation and stress relief after consumption [108]. It measures the level of positive emotions a product is likely to evoke in consumers. The aesthetic value of visual effects is determined by how consumers perceive and respond to them [109]. This refers to the appearance and packaging of takeout food. Prestige value refers to the feelings of higher status and social status felt by consumers during and following consumption of the food [110]. One of the factors that contribute to prestige value is the attitude consumers have at the moment of purchase. A transaction's value refers to the excitement and pleasure that consumers experience when they believe they have made a wise purchase [111]. A consumer's hedonic value is a measure of the level of enjoyment they receive from their purchase [112]. Based on the results of food display, we concluded that the words picture and box in the topic correspond to the results. Attitude and telephone may be associated with prestige. Additionally, words such as remaining taste, diarrhea, cost performance, cola, hair, unpalatable, etc., seem to have an impact on the evaluation results of two or more dimensions. Accordingly, the vocabulary in Topic 3 refers to the concepts of functional value, emotional value, and social value [113].

### 4.2. Study 2—The Differences between Traditional Packaging and AR Packaging

This stage of the research examines whether the use of augmented reality technology in takeout packaging will influence consumer preferences. In other words, to determine whether there is a difference in consumer evaluation results when augmented reality packaging is compared to traditional packaging. MANOVA is used to determine if there are significant differences between consumers' perceptions of trust, satisfaction, and purchase intention for two types of packaged takeaways, in order to test the effectiveness of augmented reality technology in improving consumer negative evaluations.

The reliability of these constructs was assessed through Cronbach' $\alpha$ testing after the survey was conducted and questionnaires were returned. According to the results: the construct reliability of trust $\alpha = 0.738$, the construct reliability of satisfaction $\alpha = 0.845$, and the construct reliability of purchase intention $\alpha = 0.863$. Each construct has a Cronbach's $\alpha$ coefficient greater than 0.7. As a result, the new reliability after deleting any item is lower than the original reliability. In conclusion, the reliability of each construct item is lower than its original reliability [114].

We used Levene's and Box's tests to determine whether the data distribution meets the conditions for multivariate analysis of variance. According to the results, the leaven statistic for each construct is less than 1.96, and the significance level is greater than 0.05. Thus, sample variances and population variances are not statistically different [115]. In multiple variance–covariance matrices, the significance of Box's M is greater than 0.05, which indicates that variances and covariances are equal [116]. Overall, the data are consistent with the assumption of homogeneity and are suitable for further investigation.

In Table 3, the results of the multivariate analysis of variance are presented. The control group is represented by group I, which represents the results of the consumer evaluation of traditional takeout containers. Group J represents the experimental group, which represents the evaluation results of consumers concerning augmented reality packaging. A significant difference was found between the experimental and control groups in terms of trust, satisfaction, and purchase intention evaluations (F > 1.96, $p < 0.05$). Similarly, according to Richardson [117], satisfaction and trust have small effect sizes (partial $\eta2 > 0.099$), and the test results of purchase intention have a medium effect size (partial $\eta2 > 0.059$). Overall, it appears that takeaway packaging can affect consumer perception during the retail process. Consumers' trust, satisfaction, and purchase intention have been shown to be higher with augmented reality packaging than with traditional packaging in the retail process.

**Table 3.** Multiple comparisons.

| Construct | (I) Group | (I) Mean | (J) Group | (J) Mean | Mean Difference (I-J) | F | Sig. | Partial Eta Squared |
|---|---|---|---|---|---|---|---|---|
| TR | 1 | 3.011 | 2 | 3.302 | 0.291 | 4.862 | 0.030 * | 0.045 |
| SA | 1 | 2.849 | 2 | 3.235 | 0.386 | 4.891 | 0.029 * | 0.046 |
| PuI | 1 | 2.522 | 2 | 3.072 | 0.550 | 8.910 | 0.004 * | 0.080 |

* The level of significance is 0.05.

### 4.3. Study 3—The Effectiveness of AR Packaging in Improving Negative Evaluations

We conducted quantitative research to establish the relationship between augmented reality technology and consumers' negative evaluations of takeaway packaging through the use of the technology incentive model. Each construct can pass the test of reliability and validity after deleting NE1, FL1, QoS3, TR1, and TR5. We therefore delete the above items and calculate the data, and the results are as follows.

### 4.3.1. Reliability Analysis

The results of the reliability analysis show that the corrected item total correlation for each construct is greater than 0.5, indicating that items within each construct have similar scores [118]. Additionally, each construct has a reliability greater than 0.7, and deleting any item does not produce a higher reliability. Overall, the data are reliable and suitable for further analysis.

### 4.3.2. Exploratory Factor Analysis

Through exploratory factor analysis, we assess the single-construct nature of the data. Principal component analysis is selected as the calculation method, and Varimax is used to rotate the axis. We conducted Kaiser–Meyer–Olkin (KMO) and Bartlett's sphere tests in order to meet the prerequisites for exploratory factor analysis. According to the results, all KMO values are greater than 0.5, and Bartlett's sphere test is less than 0.05 significant.

As a result, the correlation matrix suggests a partial correlation between items, and the null hypothesis that the correlation matrix is an identity matrix is rejected. In this regard, it is suitable for exploratory factor analysis [119]. In addition, we analyze the results of extracting new factors with eigenvalues greater than 1 for each construct. We find that there can only be one new factor generated from all constructs, with a total variance explained of greater than 60%. Thus, the new factors can explain the original items well, as there are no multiple sub-concepts within the construct. Currently, the commonality of all items is greater than 0.5, and the factor loading is greater than 0.6. In light of this finding, there is a correlation between items that are within the same construct, meeting the indicators suggested in previous studies [120]. Overall, we believe that the survey results adequately demonstrate the one-dimensional nature of the issue.

### 4.3.3. Confirmatory Factor Analysis

Using confirmatory factor analysis, we tested the convergent validity and discriminant validity of each construct. Additionally, we calculated the common latent factor method (CCLFM) to establish a control model to test the common method bias. The fitting results of the model are shown in Table 4. All fitting indicators of the confirmatory factor analysis met the recommended standards in previous studies, which indicates that the model fits well [121]. Additionally, we found that there was no significant change in indexes between CCLFM and CFA based on the results of model fitting. Therefore, compared to CFA, CCLFM reduces RMSEA, but the reduction is less than 0.05, and the results of improving GFI, AGFI, NFI, and CFI are less than 0.1. In terms of SRMR indicators, CCLFM has actually produced better fitting results. Based on the comparison of the fitting results, we found that there is no apparent method bias in the data [122].

**Table 4.** Measures of fit for CFA and CCLFM.

| Common Indices | $\chi^2$/df | RMSEA | GFI | AGFI | NFI | CFI | SRMR |
|---|---|---|---|---|---|---|---|
| Judgment criteria | <5 | <0.08 | >0.9 | >0.9 | >0.9 | >0.9 | <0.08 |
| CFA Value | 1.996 | 0.032 | 0.950 | 0.936 | 0.956 | 0.977 | 0.027 |
| CCLFM Value | 1.952 | 0.031 | 0.952 | 0.939 | 0.957 | 0.978 | 0.031 |

The results of the confirmatory factor analysis are shown in Table 5. The results show that all items had a factor loading greater than 0.6, and the squared multiple correlation (SMC) was greater than 0.4, meeting previous standards [123]. According to our calculations, our average variance extracted (AVE) for each construct is greater than 0.5, indicating that all items in each construct explain the construct in the same manner [124]. Additionally, the composite reliability (CR) test results are all greater than 0.7. Overall, we believe that the constructs of the model have convergent validity.

We used the Fornell–Larcker criterion method to test the discriminant validity between the constructs, and the results are shown in Table 6 [125]. Based on this method, the square root value of AVE is calculated and compared with the Pearson correlation coefficient between the two constructs. In the present study, the square root of AVE for each construct is greater than its correlation coefficient with other constructs. Therefore, we believe that the constructs have good discriminant validity.

### 4.3.4. Structural Equation Model (SEM)

To verify the hypothesized model, we established a structural equation model using AMOS. During the calculation process, maximum likelihood was used, bootstrap was run 2000 times, and a 95% confidence interval was established. According to Table 7, all model fitting indicators meet the recommended standards [121].

**Table 5.** Results of the convergent validity.

| Construct | Coding | Factor Loading | t Value | SE | *p* Value | SMC | AVE | CR |
|---|---|---|---|---|---|---|---|---|
| PI | PI1 | 0.718 | 23.911 | 0.089 | 0.001 * | 0.516 | | |
| | PI2 | 0.679 | 22.251 | 0.106 | 0.001 * | 0.460 | 0.530 | 0.771 |
| | PI3 | 0.782 | 26.633 | 0.098 | 0.001 * | 0.611 | | |
| PV | PV1 | 0.732 | 24.415 | 0.095 | 0.001 * | 0.535 | | |
| | PV2 | 0.742 | 24.849 | 0.094 | 0.001 * | 0.550 | 0.554 | 0.789 |
| | PV3 | 0.759 | 25.596 | 0.100 | 0.001 * | 0.576 | | |
| NE | NE2 | 0.806 | 28.730 | 0.093 | 0.001 * | 0.649 | | |
| | NE3 | 0.821 | 29.494 | 0.100 | 0.001 * | 0.675 | 0.654 | 0.850 |
| | NE4 | 0.799 | 28.392 | 0.103 | 0.001 * | 0.638 | | |
| FL | FL2 | 0.811 | 29.721 | 0.104 | 0.001 * | 0.657 | | |
| | FL3 | 0.863 | 32.574 | 0.098 | 0.001 * | 0.745 | 0.696 | 0.873 |
| | FL4 | 0.828 | 30.615 | 0.104 | 0.001 * | 0.685 | | |
| QoS | QoS1 | 0.720 | 24.135 | 0.091 | 0.001 * | 0.519 | | |
| | QoS2 | 0.705 | 23.479 | 0.101 | 0.001 * | 0.497 | 0.575 | 0.801 |
| | QoS4 | 0.843 | 28.090 | 0.093 | 0.001 * | 0.657 | | |
| QoF | QoF1 | 0.830 | 31.265 | 0.093 | 0.001 * | 0.690 | | |
| | QoF2 | 0.795 | 29.301 | 0.094 | 0.001 * | 0.633 | 0.673 | 0.892 |
| | QoF3 | 0.843 | 32.019 | 0.093 | 0.001 * | 0.711 | | |
| | QoF4 | 0.812 | 28.090 | 0.099 | 0.001 * | 0.659 | | |
| PVL | PVL1 | 0.823 | 29.952 | 0.097 | 0.001 * | 0.677 | | |
| | PVL2 | 0.772 | 27.398 | 0.095 | 0.001 * | 0.597 | 0.634 | 0.839 |
| | PVL3 | 0.793 | 28.429 | 0.100 | 0.001 * | 0.629 | | |
| TR | TR2 | 0.780 | 27.330 | 0.094 | 0.001 * | 0.608 | | |
| | TR3 | 0.746 | 25.716 | 0.098 | 0.001 * | 0.556 | 0.588 | 0.811 |
| | TR4 | 0.774 | 27.030 | 0.094 | 0.001 * | 0.598 | | |
| SA | SA1 | 0.804 | 29.186 | 0.094 | 0.001 * | 0.647 | | |
| | SA2 | 0.773 | 27.609 | 0.092 | 0.001 * | 0.598 | 0.617 | 0.829 |
| | SA3 | 0.780 | 27.932 | 0.091 | 0.001 * | 0.608 | | |
| PuI | PuI1 | 0.836 | 31.293 | 0.099 | 0.001 * | 0.699 | | |
| | PuI2 | 0.833 | 31.115 | 0.102 | 0.001 * | 0.694 | 0.690 | 0.870 |
| | PuI3 | 0.823 | 30.560 | 0.102 | 0.001 * | 0.678 | | |

\* The level of significance is 0.05.

**Table 6.** Test results for discriminant validity.

| | PI | PV | NE | FL | QoS | QoF | PVL | TR | SA | PuI |
|---|---|---|---|---|---|---|---|---|---|---|
| PI | 0.738 | | | | | | | | | |
| PV | 0.545 * | 0.744 | | | | | | | | |
| NE | 0.366 * | 0.286 * | 0.809 | | | | | | | |
| FL | 0.399 * | 0.364 * | 0.470 * | 0.834 | | | | | | |
| QoS | 0.454 * | 0.380 * | 0.365 * | 0.444 * | 0.758 | | | | | |
| QoF | 0.421 * | 0.415 * | 0.381 * | 0.462 * | 0.517 * | 0.820 | | | | |
| PVL | 0.457 * | 0.433 * | 0.346 * | 0.453 * | 0.470 * | 0.593 * | 0.796 | | | |
| TR | 0.408 * | 0.312 * | 0.352 * | 0.487 * | 0.520 * | 0.574 * | 0.530 * | 0.767 | | |
| SA | 0.512 * | 0.476 * | 0.328 * | 0.479 * | 0.496 * | 0.545 * | 0.596 * | 0.549 * | 0.785 | |
| PuI | 0.506 * | 0.439 * | 0.357 * | 0.484 * | 0.456 * | 0.530 * | 0.540 * | 0.533 * | 0.682 * | 0.831 |

\* The level of significance is 0.05.

**Table 7.** Adaptability of SEM.

| Common Indices | $\chi^2$/df | RMSEA | GFI | AGFI | NFI | CFI | SRMR |
|---|---|---|---|---|---|---|---|
| Judgment criteria | <5 | <0.08 | >0.9 | >0.9 | >0.9 | >0.9 | <0.08 |
| Value | 2.701 | 0.041 | 0.928 | 0.914 | 0.936 | 0.958 | 0.047 |

As shown in Figure 8, a structural model is presented. In the figure, we have marked the degree of influence. According to previous studies, path coefficient values ranging from

0.1 to 0.3 represent weak influence levels, 0.3 to 0.5 represent neutral influence levels, and 0.5 to 1.0 represent strong influence levels [126]. A significant path relationship between flow and trust is the only path that is significant in the model. The model fitting results indicate that each construct has a positive relationship with the others.

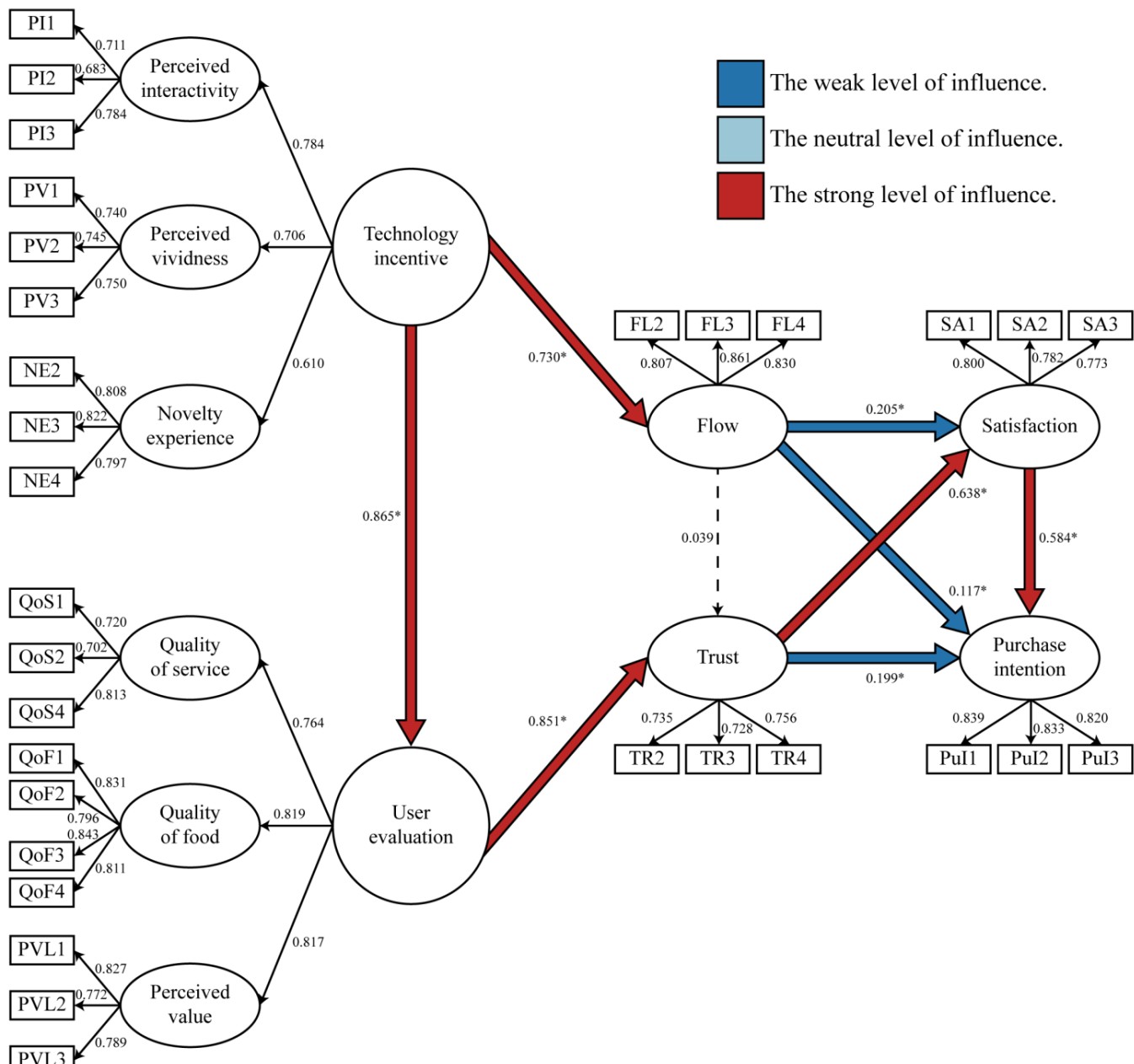

**Figure 8.** Results of the structural equation model. * The level of significance is 0.05.

In our study, we examine the direct effect, indirect effect, and total effect between constructs. Table 8 presents the results. According to our findings, technology incentives have a significant direct effect on both user evaluation and flow experience ($p < 0.05$). The validity of H1a and H2a has been established. It has been pointed out by McLean and Wilson [46] that perceived interactivity, perceived vividness, and novelty experience are the main factors that will influence consumers' experience of augmented reality. In this study, the second-order dimension technology incentive constituted by them is implicated to some extent in improving consumers' negative comments on food delivery through augmented reality.

**Table 8.** Results of the Path Analysis Test.

| Hypothesis | Path | Direct Effect | | Indirect Effect | | Total Effect | | Results |
|---|---|---|---|---|---|---|---|---|
| | | β | B-C Sig. | β | B-C Sig. | β | B-C Sig. | |
| H1a | TI→UE | 0.865 | 0.001 * | / | / | 0.865 | 0.001 * | Support |
| H2a | TI→FL | 0.730 | 0.001 * | / | / | 0.730 | 0.001 * | Support |
| | TI→TR | / | / | 0.764 | 0.001 * | 0.764 | 0.001 * | |
| | TI→SA | / | / | 0.637 | 0.002 * | 0.637 | 0.002 * | |
| | TI→PuI | / | / | 0.610 | 0.001 * | 0.610 | 0.001 * | |
| H3a | UE→TR | 0.851 | 0.001 * | / | / | 0.851 | 0.001 * | Support |
| | UE→SA | / | / | 0.542 | 0.001 * | 0.542 | 0.001 * | |
| | UE→PuI | / | / | 0.487 | 0.001 * | 0.487 | 0.001 * | |
| H4a | FL→TR | 0.039 | 0.484 | / | / | 0.039 | 0.484 | Not support |
| H5a | FL→SA | 0.205 | 0.005 * | 0.025 | 0.484 | 0.230 | 0.001 * | Support |
| H6a | TR→SA | 0.638 | 0.001 * | / | / | 0.638 | 0.001 * | Support |
| H7a | FL→PuI | 0.117 | 0.001 * | 0.142 | 0.001 * | 0.258 | 0.001 * | Support |
| H8a | TR→PuI | 0.199 | 0.004 * | 0.373 | 0.001 * | 0.572 | 0.001 * | Support |
| H9a | SA→PuI | 0.584 | 0.001 * | / | / | 0.584 | 0.001 * | Support |

* The level of significance is 0.05.

Our study examines two factors that may affect consumer trust, namely, user evaluation and flow. According to the results, user evaluation has a significant direct impact on trust ($p < 0.05$), but the path coefficient between flow and trust is not significant ($p > 0.05$). Therefore, H3a is proved to be true, whereas H4a is rejected. The results of this study support a proposition made in e-commerce research [127] that the stimuli generated by online comments can have an effect on consumer trust. While there is a relationship between flow and trust in the technology incentive model [16], our survey results indicate that setting up this relationship is affected by both the research field and the research objects.

Additionally, we examine the direct effects of flow and trust on consumer satisfaction. Results indicate that these two dimensions play an important role in determining satisfaction with the workplace ($p < 0.05$). The validity of H5a and H6a is therefore established. Obtaining flow experience will enhance the viewing experience and satisfaction of spectators of sports events using virtual reality technology [47]. We found that consumer satisfaction in the field of food delivery is also influenced by flow and trust.

Our study found that flow, trust, and satisfaction have a direct impact on purchase intent ($p < 0.05$). The validity of H7a, H8a, and H9a is therefore established. The relationship between these aspects of online retailing has been discussed in several studies and preliminary path relationships have been proposed [128]. We provide further evidence that elevating consumer perceptions can lead to higher food delivery sales by elevating consumer perceptions.

Further, the path analysis results indicate that technology incentives have a significant indirect effect on trust, satisfaction, and purchase intentions ($p < 0.05$). In addition, user evaluation has a significant indirect effect on satisfaction and purchase intent ($p < 0.05$). Therefore, flow, trust, and satisfaction all play an important role as intermediary variables. The intermediary role of these constructs may be the reason for the impact of technology incentive and user evaluation on sales. This is also an area of focus in marketing.

Moreover, we examined the moderating effects of three variables on the path relationship between constructs, namely, gender, age, and experience with augmented reality. The data is presented in Table 9. In our study, gender has a significant moderating effect only on the impact path of technology incentive on flow; there is a significant difference between the hypothetical model (male β= female β) and the original model. The experience only modifies the impact path of technology incentives on flow and the impact path of trust on satisfaction, i.e., there is a significant difference between the hypothetical model (yes β = no β) and the original model. In all path relationships, age does not have a significant moderating effect. In light of this, only three assumptions were supported

regarding moderating effects: gender on the path of technology incentive and flow (H2b), experience on the path of technology incentive and flow (H2d), and experience on the path of trust and satisfaction (H6d).

**Table 9.** Results of the moderation effect.

| Path | Gender | | Age | | Experience | |
|---|---|---|---|---|---|---|
| | CMIN | *p* | CMIN | *p* | CMIN | *p* |
| TI→UE | 2.295 | 0.130 | 0.163 | 0.687 | 0.776 | 0.378 |
| TI→FL | 7.435 | 0.006 * | 0.246 | 0.620 | 8.523 | 0.004 * |
| UE→TR | 1.286 | 0.257 | 1.001 | 0.317 | 0.574 | 0.449 |
| FL→TR | 0.259 | 0.611 | 0.269 | 0.604 | 0.520 | 0.471 |
| FL→SA | 0.003 | 0.957 | 0.219 | 0.640 | 2.822 | 0.093 |
| FL→PuI | 0.005 | 0.942 | 1.343 | 0.247 | 0.184 | 0.668 |
| TR→SA | 3.080 | 0.079 | 0.940 | 0.332 | 10.781 | 0.001 * |
| TR→PuI | 1.635 | 0.201 | 1.410 | 0.235 | 1.797 | 0.180 |
| SA→PuI | 3.403 | 0.065 | 1.939 | 0.164 | 0.025 | 0.874 |

* The level of significance is 0.05.

According to Table 10, we calculated and compared the specific path coefficients of the influence paths with significant regulation effects. In terms of the effect of technology incentives on flow, women are significantly higher than men, based on these results from a gender perspective. As a result, women have a greater likelihood of experiencing a flow experience since the package contains augmented reality technology.

**Table 10.** The comparison of path coefficients with significant moderating effects.

| Moderating Variable | | Path | β | *p* |
|---|---|---|---|---|
| gender | male | TI→FL | 0.709 | 0.001 * |
| | female | | 0.757 | 0.001 * |
| experience | yes | TI→FL | 0.749 | 0.001 * |
| | no | | 0.550 | 0.001 * |
| | yes | TR→SA | 0.714 | 0.001 * |
| | no | | 0.442 | 0.003 * |

* The level of significance is 0.05.

In terms of experience, consumers with augmented reality interaction experience have greater path coefficients than consumers without such experience in the relationship between technology incentive and flow, as well as the relationship between trust and satisfaction. Hence, the interactive experience of augmented reality makes it easier for consumers to enjoy interactive products and for businesses to achieve their marketing goals.

## 5. Discussion

There is significant potential for augmented reality marketing to contribute to society as a whole, whether it is for profit or not for profit [21]. In this study, we provide design suggestions for promoting the commercial application of augmented reality technology, as well as broadening the usage of augmented reality for marketing purposes. Furthermore, the COVID-19 pandemic is impacting the offline dining environment of many physical restaurants [129]. To cope with the epidemic, some operators have accelerated the transition from traditional sales to takeaway [130]. On the other hand, judging the behavior of takeaway consumers accurately may require additional theoretical foundations and practical business experience, unlike offline dining. We provide catering operators with a reference for formulating marketing strategies from the perspective of user evaluation based on our research findings. In Table 11, we summarize the content and results of the three phases of research.

**Table 11.** Summary of survey results.

| Studies | Content | Constructs | Results |
|---|---|---|---|
| Study 1 | The purpose of this study is to collect negative evaluations from food delivery platforms for using Latent Dirichlet Allocation topic modeling to identify the factors that affect the evaluation of users. | QoS, QoF, PVL | Based on the text analysis model, low levels of QoS, QoF, and PVL are most likely to result in negative consumer evaluations. |
| Study 2 | Comparing augmented reality packaging with traditional takeaway packaging in the design of takeaway packaging. | TR, SA, PuI | Users' evaluation of the TR, SA, and PuI of takeaway packaged in augmented reality is significantly higher than that of takeaway packaged in traditional packaging. The second-order construct TI composed of PI, PI, and NE may enhance the second-order construct UE consisting of QoS, QoF, and PVL. Finally, increasing the PuI of consumers through FL, TR, SA intermediaries. |
| Study 3 | A structural equation model was developed to analyze the mechanism of augmented reality technology in takeout packaging in order to improve consumer negative evaluations. | PI, PV, NE, TI, FL, QoS, QoF, PVL, UE, TR, SA, PuI | |

It has been found that if food and beverage outlets that provide takeaway services fail to satisfy consumers sufficiently in terms of survey quality, food quality, and perceived value, they are very likely to receive negative feedback from customers. These three concepts are considered to be antecedents of user evaluation. In the past, a consumer's negative perception of a restaurant might have only a limited impact on those closest to him. It is important to note that the impact of negative evaluations is extremely limited. Due to the massive popularity of social media, many consumers are willing to share their dining comments with others on the website or in the community [131]. Therefore, while the traditional catering industry can benefit from the opportunities brought about by computerization, it may also face new challenges as a result of digitization. From another perspective, this also illustrates the importance of modern catering marketing, which emphasizes user evaluation antecedents. We have been able to clearly define these three concepts in the food delivery industry through text analysis and literature review. Specifically, the quality of a survey is determined by the level of human interaction, the reasonableness of the supporting facilities, and the quality of the food content. Quality of food includes freshness, health, taste, safety, and human factors. Perceived value is a comprehensive evaluation result which includes functional value, emotional value, and social value. Accordingly, we recommend that active preparation and improvement should be carried out based on the antecedents of user evaluation. The best way to avoid negative comments is to try to prevent them from occurring in the first place. According to a survey, 20% of consumers expect a response to a negative evaluation within a day, and 96% say they read a restaurant's response to a negative evaluation if the restaurant does not respond within that day [132]. In summary, the findings suggest that when consumers give negative comments about takeaway food, the restaurant can detect the possible reasons in time and analyze the probable causes based on the antecedents of the evaluation; then, replying with high quality as soon as possible is an effective way to minimize the negative impact of these comments.

Furthermore, our findings confirm the importance of utilizing augmented reality technology in the design of takeaway food packaging. The three dimensions by which we evaluate consumers' perceptions are trust, satisfaction, and intention to purchase.

Surprisingly, consumers' evaluation of takeaway packaging utilizing augmented reality technology appears to be positive across all aspects. In other words, consumers are more likely to choose takeaway packages bundled with augmented reality technology whether it is due to trust, satisfaction, or purchase intent. With the continuous development of new technologies such as mobile augmented reality applications, some studies suggest that retailers can improve the experience and value for consumers [133]. It is possible for retailers to use augmented reality technology to improve their service offerings and receive more positive responses from consumers using this technology [134]. This study confirms that augmented reality can make an important contribution to takeaway sales and packaging design in the takeaway sector. In some arbitrary and intuitive views, the takeaway packaging is more important for functions such as heat preservation and spill prevention. The importance of visual and interactive experiences seems to be declining, as consumers are more inclined to engage in activities after eating [135]. Our study questions this claim. Takeaway packaging should also consider the consumer's visual experience as well as their interactive experience. Based on the findings of the study, it is possible to speculate that some consumers are attracted to augmented reality technology before dining in order to receive a high-quality interactive experience. During the meal, positive emotions probably reinforce overall evaluations of the food and establishment. Despite the fact that another group of consumers prefers to consume food immediately rather than interact with the packaging, they may also be attracted to augmented reality packaging after eating the food and interact with it in that manner. The use of augmented reality technology may improve the perceptions and preferences of these consumers after meals.

　　Lastly, based on the technology incentive model, we verify and analyze the mechanism that allows augmented reality packaging to enhance marketing. Our research results contribute to the development of a theoretical model of food retailing as a result of our research. A secondary dimension of technology incentive provided by augmented reality technology is perception of interaction, perceived vividness, and novelty [16]. This can have a positive effect on consumer evaluation in catering marketing. This is similar to Tang and Chang [136], who used the decision tree algorithm of machine learning to model and analyze consumer behavior regarding food delivery; they found that user evaluation and satisfaction are closely related. One of the ways that new technologies may help marketing is to improve user evaluation. Based on the theoretical model we developed, it is clearly demonstrated that the characteristics of interactive experiences based on augmented reality, such as interactivity, vividness, and novelty, affect user evaluations of services, food, and perceived value. It is true that the ultimate goal of retail is to facilitate purchases. However, factors such as flow experience, trust, and satisfaction are also aspects that catering workers strive to improve through the use of technological tools. We show in our theoretical model that augmented reality also leads to a more positive perception of these aspects by users. It should be noted that the provision of augmented reality packaging can require continuous optimization of the interaction process and continuous iteration of interactive content under adequate development conditions. The interactive experience should pay particular attention to consumer perceptions of ease of use [137]. In our moderator variable test, we found that consumers with augmented reality experience perceive the product more positively when interactive experiences are included. In contrast, novelty experience may refer to the stimulation that is brought about by a novel experience. For developers and designers of augmented reality packaging, this set of concepts may appear contradictory. We suggest catering companies continuously launch new versions of augmented reality packaging and update the content of the packaging in order to maintain a high level of

consumer evaluation of the packaging's novelty. Additionally, it is important to pay attention to the coherence of interaction logic in the design, i.e., that the interaction method does not change significantly when the version changes. This makes consumers feel fresh and interactive.

## 6. Conclusions

### 6.1. Contribution

The findings of our study provide a design basis for interaction designers working in the restaurant marketing sector who want to use augmented reality in their work. Additionally, based on the text analysis of consumer evaluations and the path relationship of the established quantitative model, we have developed the key points of business strategies for catering businesses. In contrast to previous investigations of food marketing, this study examines the content of user comments and considers how the interactive experience can be beneficial. It has been shown that the technology incentive of augmented reality packaging can be effective in improving consumers' negative comments. Furthermore, it can also enhance consumers' willingness to purchase under the intermediary of flow, trust, and satisfaction.

Specifically, restaurants with a large number of negative consumer evaluations should consider paying close attention to packaging design, especially using augmented reality technology to address their issues. The results of our study demonstrate that interactive packaging can improve consumers' negative perceptions of takeaway food and improve their evaluations of it. Moreover, this improvement encompasses a variety of aspects, not only addressing insufficient packaging design as a source of negative evaluations. While the results of the comparison of building models based on gender, age, and whether users have interactive experience using augmented reality show that classification characteristics of consumers do not cause changes in consumer behavior in most cases, women and consumers with more interactive experience still responded positively to augmented reality marketing strategies. Thus, restaurants that offer takeaway services should first analyze their own consumer characteristics. It will be more likely to consider the actual management strategy of using AR in packaging to minimize negative consumer feedback in the event that a significant portion of the main consumer group matches the characteristics of women and AR interactive experience.

Our theoretical contributions include the proposed reasons for consumers' negative reviews based on the results of text analysis performed using unsupervised machine learning methods, along with the development of a theoretical framework based on the technology incentive model. A verified version of our new theoretical model is shown in Figure 9, and we refer to it as the expanded technology incentive model (ETIM). The theoretical model we propose may be applicable to human–computer interactions in broader fields of food marketing, and is not limited to augmented reality applications. The framework examines three main characteristics of interactive products, namely, perceived interactivity, perceived vividness, and novelty experience, and evaluates the impact of this experience on consumer behavior [16]. As shown by our verification results, augmented reality technology is an effective strategy for restaurants that offer takeaway food. The perception of interactivity, perceived vividness, and novelty experience all contribute to marketing success. While it has been established that there is no direct and significant relationship between consumers' flow experience and trust, the incentive experience of human–computer interaction has been shown to improve consumers' antecedents of evaluation in order to ultimately increase the effect of purchase intention.

### 6.2. Limitions and Future Studies

To begin with, all the survey objects are from China, and are primarily young people between the ages of 20 and 40. Neither younger nor older consumers were surveyed [96]. When choosing to buy takeaway food, consumers in single-person households value variety the most, while consumers in multi-person households place a greater emphasis

on design. A difference in living conditions may result in differences in consumer behavior. A difference in personal preferences may also result from a difference in living conditions.We therefore propose that further research be conducted in order to develop a detailed classification based upon the characteristics and lifestyles of consumers, as well as to obtain more specific marketing and design references.

Furthermore, packaging and study samples were chosen without regard to the way the restaurant's food is prepared, how it tastes, or the company's image. One of the factors affecting consumers may be the level of cooking or the brand of the restaurant, and among the consumers surveyed, some have had similar dining experiences and others have not. Therefore, this is one of the limitations of this study. Future research can broaden the scope of the study, sample from major cuisines, and select the most representative research samples based on the restaurant's operating conditions.

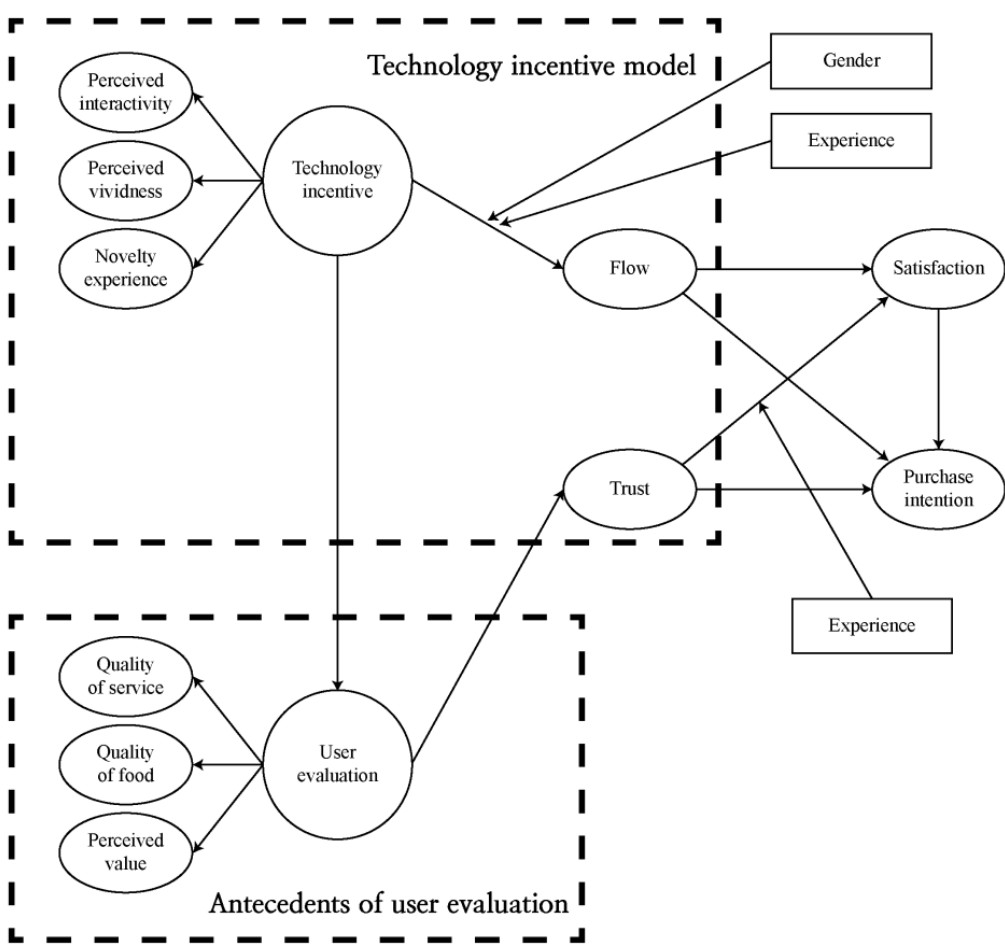

**Figure 9.** Expanded technology incentive model.

Thirdly, our research focuses on the process by which negative evaluations are formed and how to improve the negative evaluations of consumers. Nevertheless, positive evaluation may be one of the research topics that requires further study. We chose not to use positive evaluations because in the current Internet catering industry, there are many reviewers hired by merchants to post false positive evaluations. Alternatively, negative comments are more authentic and are more likely to reflect the feelings of actual users. Accordingly, we suggest that researchers in future studies analyze the reasons for consumers'

positive evaluations in order to distinguish between true and false positive evaluations. An analysis of positive and negative sentiment in integrated texts will assist in jointly developing marketing strategies for food.

In addition to solving the three problems mentioned above, the comparison of cost increases and return benefits associated with augmented reality packaging is also one of the scopes that can be examined for future research. It is inevitable that the adoption of augmented reality technology will increase the design costs during the process of in-depth development and iteration [138]. It is worth evaluating whether restaurants can reduce unit price costs through a large number of sales. Fortunately, unlike other interactive devices, augmented reality technology supports mobile phones and tablets, which also helps to reduce the hardware cost [139]. Catering companies that have been developing augmented reality packaging for a considerable period of time may be able to lower the cost of a single augmented reality food package to a level similar to that of traditional packaging. It is, however, necessary to conduct further evaluations. In addition, packaging sustainability is an important issue that should be taken into consideration [140]. With augmented reality technology, there is no additional hardware requirement, which makes it a more environmentally friendly and sustainable alternative to other complex interactive experiences. The user must, however, actively download the software launched by the restaurant before engaging in any interaction. It is possible for some users to express distrust toward third-party software [141]. Furthermore, users may be unwilling to download additional software as they are concerned about their mobile device's capacity [142]. In addition, this indicates that guiding users to download software launched by catering brands, and maintaining their willingness to continue using these products, are important marketing issues that should be investigated further.

**Author Contributions:** Conceptualization, C.G.; methodology, S.L. and H.S.; software, W.W. and T.H.; validation, W.M. and H.S.; formal analysis, J.S.; investigation, J.C.; data curation, W.W. and W.M.; writing—original draft preparation, C.G. and T.H.; writing—review and editing, C.Y.; visualization, J.C. and W.M.; supervision, S.L.; project administration, C.Y. and J.S. All authors have read and agreed to the published version of the manuscript.

**Funding:** This research received no external funding.

**Institutional Review Board Statement:** This study was conducted according to the guidelines of the Declaration of Helsinki and received academic ethics review and approval from the review committee of the Ministry of Social Science, Changshu Institute of Technology. In our experiments, informed consent was obtained from all participants and all methods were performed per relevant guidelines and regulations.

**Informed Consent Statement:** This study does not involve disease treatment or patients, nor does it involve subjects who can be identified. Informed consent was obtained from all subjects involved in the study.

**Data Availability Statement:** The data that support the findings of this study are available from the corresponding author upon reasonable request.

**Acknowledgments:** We are grateful to Liao Jiang for their work in conceptualization. We also thank the anonymous reviewers who provided valuable comments on the manuscript.

**Conflicts of Interest:** The authors declare no conflict of interest.

## Appendix A

The items used for the MANOVA and SEM are shown in Table A1.

**Table A1.** The scale of measurement.

| Constructs | Items | Source |
| --- | --- | --- |
| Perceived interactivity | PI1: Overall, using AR takeaway packaging is highly interactive.<br>PI2: The interaction in takeaway packaging using AR technology is efficient and clear.<br>PI3: Takeaway packaging using AR technology fits well with my needs. | [143] |
| Perceived vividness | PV1: Using AR takeaway packaging makes me feel dynamic.<br>PV2: Using AR takeaway packaging provides me with a lot of vividness.<br>PV3: I enjoy using AR takeaway packaging. | [144] |
| Novel experience | NE1: Many aspects of takeaway packaging using AR technology were novel to me. (Deleted)<br>NE2: Using AR takeaway packaging provided a unique experience for me.<br>NE3: Using AR takeaway packaging has been an adventurous experience.<br>NE4: I felt I was in a different world during using AR takeaway packaging. | [16] |
| Flow | FL1: When using AR takeaway packaging, I am not distracted. (Deleted)<br>FL2: It feels like time flies while I am using AR takeaway packaging.<br>FL3: When using AR takeaway packaging, I have a feeling of concentration.<br>FL4: When using AR takeaway packaging, I don't get distracted from other things. | [145] |
| Quality of service | QoS1: I think using AR takeaway packaging makes me feel concern and help.<br>QoS2: I think the AR takeaway packaging got me a quick response.<br>QoS3: I think the AR takeaway packaging offers various services. (Deleted)<br>QoS4: I think using AR takeaway packaging is respectful for customers. | [104] |
| Quality of food | QoF1: I think the AR takeaway packaging helps the food to have a good presentation.<br>QoF2: I think the AR takeaway packaging helps make the food look more various.<br>QoF3: I think the AR takeaway packaging helps make the food look and taste better.<br>QoF4: I think the AR takeaway packaging helps make the food look healthier. | [99] |
| Perceived value | PVL1: I think takeaways that use AR packaging are great value for the price.<br>PVL2: The overall value of eating takeaways that use AR packaging is high.<br>PVL3: Using AR takeaway packaging was worth the money. | [102] |
| Trust | TR1: I can count on this company/companies that use AR takeaway packaging to consider how their actions will affect customers like me. (Deleted)<br>TR2: If I were to have any problems, this company/companies that use AR takeaway packaging will be ready and willing to offer me assistance and support.<br>TR3: When making decisions about its policies, this company/companies that use AR takeaway packaging is concerned about customers like me.<br>TR4: I can count on this company/companies that use AR takeaway packaging to be sincere in its communication.<br>TR5: Even if this company/companies that use AR takeaway packaging were to provide an unlikely explanation, I would be confident that the explanation was correct. (Deleted) | [146] |
| Satisfaction | SA1: I am satisfied with takeaways of this company/companies that use AR takeaway packaging.<br>SA2: Considering all my experience with food, my choice of takeaway of this company/companies that use AR takeaway packaging was wise.<br>SA3: Overall, I am pleased with takeaway of this company/companies that use AR takeaway packaging based on my experience. | [147] |
| Purchase intention | PuI1: I regard this company/companies that use AR takeaway packaging as my first choice for takeaway purchases.<br>PuI2: I plan to order more takeaways from of this company/companies that use AR takeaway packaging in the next few years.<br>PuI3: I will continue to buy takeaways at of this company/companies that use AR takeaway packaging in the next few years. | [146] |

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
