# Peer review of "The Effect of Using Augmented Reality Technology in Takeaway Food Packaging to Improve Young Consumers’ Negative Evaluations"

_agriculture, doi:10.3390/agriculture13020335_

Round 1

Reviewer 1 Report

I would like to start by complimenting the authors for both the choice of the topic (in the context of its relevance and novelty) and the manner in which they deal with it (by conducting three studies and relying on a wide literature). In addition, the style is very clear and it is evident that the authors know what they want to say.

Nevertheless, I feel the need to address some issues that I believe need to be resolved before I would suggest with no doubts for the paper be published. I will roughly divide them into two topics:

1. Literature review and interpretation

Generally, the authors deal with several mutually connected topics: packaging design, augmented reality, and consumer evaluations. All the topics are considered in the paper, but in its different parts – Introduction/Literature review/Study 1 results.  

Within the literature review authors dominantly focus on variables from studies 2 and 3.

(i) I suggest that all the topics need to be addressed in this part of the paper. Therefore, I recommend adding a new section that would generally deal more deeply with the influence of evaluations provided on the Internet on consumer behavior as well as corresponding methodological approaches.

(ii) In addition, the influence of packaging design on consumer choice should be also generally considered at this place.

(iii) On the other hand, when providing the results of study 1, the authors should emphasize packaging design more since it is the factor treated in another two studies.

(iv) Finally, since implementing moderation analysis, the authors should formulate appropriate hypotheses and provide literature background for them as well.

 2. Methodology:

(i) The sample – the authors should add (in the part of the text in which they present characteristics of the sample of respondents) the type of the sample and sampling frame.

(ii) The design of the second study – there are two issues that need to be resolved in this context:

(a) authors need to specify when selecting “ 3 national chain catering brands that currently offer food delivery services“ on the basis of “ a large number of negative evaluations on the food delivery platform” whether those negative evaluations were in regard to their packaging design or generally.

(b) Since during conducting study 2 there were “two measurements being repeated on each subject” whereby “the subjects completed the questionnaire for the original packaging, then observed the augmented reality packaging and completed the second questionnaire” it remains unclear how they provided answers for the three constructs considering original packaging when all the items for these constructs provided in appendix A are formulated from the context of implementing AR.

(iii) The relationships between first-order constructs (PI, PV, NE) and the second-order construct (Technology incentives) – the authors should consider whether that relation is reflective or formative. In the case of the second approach, which seems more appropriate to me, the data analysis would need to be performed once again relying on, for example, SmartPLS software. When providing such discussions the authors can rely, among others, on: Jarvis, C. B., MacKenzie, S. B., Podsakoff, P. M. (2003). A Critical Review of Construct Indicators and Measurement Model Misspecification in Marketing and Consumer Research. Journal of Consumer Research, 30 (2), 199-218.

(iv) Naming of the constructs in the context of the items from the questionnaire – The authors adapted the questionnaire to AR packaging design. For example, one of the items measuring food quality is formulated in the following manner: “QoF1: I think the AR takeaway packaging helps the food to have a good presentation.” The authors should justify whether it is appropriate to name that construct as Food quality since it actually measures Food quality from the aspect of AR takeaway packaging (or perceived influence of AR takeaway packaging on food quality), and not in general. There can be found an analogy in the case of other constructs as well.

In addition to all previous comments, I could add some technical issues (for reference 21, there is not provided the name of the scientific journal; generally, links with doi numbers are missing; in row 199, there is a sign I believe from a Chinese alphabet).

Once again, I would like to support the authors in their work.

Author Response

Response to Reviewer 1 Comments

Dear Referee,

Thank you very much for the valuable suggestions, which have made our paper more comprehensive. We have responded to each of your comments below:

Point 1: I suggest that all the topics need to be addressed in this part of the paper. Therefore, I recommend adding a new section that would generally deal more deeply with the influence of evaluations provided on the Internet on consumer behavior as well as corresponding methodological approaches.

Response 1: Thank you for the suggestion. In order to address these topics, we have added a dedicated section to section 2.4 of the literature review. The purpose of this section is to describe the impact of online reviews on consumer behavior and the strengths of the text analysis method we use. We also explain why we use this method.

Point 2: In addition, the influence of packaging design on consumer choice should be also generally considered at this place.

Response 2: Thank you for the suggestion. Added chapter 2.4 discusses in detail how packaging design influences consumer behavior, along with the progress and gaps in research in the application of augmented reality technology to packaging design. Below you will find a list of what we added.

Actual version

2.4. Consumer behavior and packaging design

     Over the past few years, with the development of technology, a growing number of platforms have enabled consumers to post reviews online. Online reviews have gained increasing attention with the development of e-commerce, since they can reflect to some extent the potential purchase opportunities for consumers[74].By reviewing consumer reviews, communicating with them, and making targeted changes in response, the manufacturer can gain a better understanding of consumers' opinions. Due to the peculiarities of online ordering, consumers cannot physically see the products they intend to purchase. It is possible for consumers who are unfamiliar with a product to effectively analyze and compare the expected results before actually making a purchase decision based on online product reviews[75].Review sites can provide valuable information regarding a product's quality, and should be considered a reliable source of information. It is especially important when it comes to products that have no previous purchasing experience or cannot be easily accessed prior to purchase[76].It is still important for consumers to read online reviews even when they are familiar with a product. Researchers have suggested that the continuous exchange of product, service, brand, company information can be accomplished through the review data on the Internet among consumers who may have purchase behaviors, consumers who are purchasing, and consumers who have purchased products[77].It can be concluded from this that consumer reviews are more than just an interaction between consumers, but also a means of communication between consumers and manufacturers. It is essential for manufacturers to be attentive to consumer comments in order to continue to make profits and improve the quality of products and services. Therefore, consumers and brands are constantly influencing one another [78].Particularly when based on online reviews, manufacturers can quickly understand consumers' thoughts and respond efficiently. Thus, in marketing-related research, it is often concluded that consumers' ability to obtain online reviews and participate in reviews may contribute to higher product satisfaction [79].It is imperative that manufacturers extract the key points of consumers' concerns from online reviews in a large number, rigorously and effectively in order to execute a more effective marketing plan. Latent Dirichlet Allocation has emerged as one of the most widely used methods for topic discovery in recent consumer behavior research, especially for non-normalized data [80].Our choice of this research method is primarily motivated by this consideration.A Latent Dirichlet Allocation topic model extracts topics from reviews using unsupervised learning methods. Among the advantages of this method is that it does not assume the grammatical attributes of the text while it is executing, and can be used effectively to identify themes in a large number of documents[81].

Once the negative comments of consumers have been understood, targeted adjustments need to be made. Furthermore, packaging design is one of the optional marketing tools. A packaging product's primary function is to protect, store, ship, sell, promote, provide service, and ensure security [82].Especially for sales and promotional purposes, packaging design can assist manufacturers in communicating the attributes of their products to consumers in an effective and efficient manner. For instance, the color of the packaging can affect consumers' perceptions of the taste of the product[83].In this regard, packaging design is an important method of conveying sensory characteristics to consumers through its visual elements. It is through these powerful and efficient means that consumers are able to effectively influence their consumption patterns[84]. Additionally, sound attributes in packaging may impact the consumer's experience and purchase decision[85].In this regard, the auditory experience is also an important component of packaging design. Studies have shown that consumers perceive brands differently based on the complexity of packaging design. The simplicity or complexity of a design express different personalities of the brand and can easily influence consumers' perceptions of it[86].An important role played by packaging design is to convey information to consumers and encourage them to purchase. Although price is certainly a non-negligible factor in influencing young people's purchasing behavior [87], design can also contribute in a sensory and cultural manner to marketing efforts in retail. Consequently, packaging design can have a significant impact on consumer behavior.

With the advancement of technology, a number of new interactive methods have shown promise in the field of marketing, such as augmented reality. It is necessary to conduct further research in the area of augmented reality as it relates to packaging images because the application effect still lacks theoretical support[88].In the food industry, there have been relatively few attempts to apply augmented reality technology. While augmented reality is widely recognized as a technology that enriches consumer, food, and environmental interactions[89].By scanning the packaging with their mobile phones, consumers are able to view and interact with virtual product models. It is possible to better understand the appearance, function, and use of a product through this feature[90].Aside from that, some information can also be conveyed by augmented reality. For example, with augmented reality, consumers and packaging can interact more easily, thus improving the traceability of food production[91].In addition, consumers will have a deeper and more comprehensive understanding of information as a result of this approach. The study found that consumers who use augmented reality packaging gain an increased understanding of food products than those who use static packaging[92].Consequently, the application of augmented reality technology in packaging can contribute to restaurants' continuous success in food marketing. However, there is still a lack of theoretical research on consumer behavior to support this interactive marketing approach. It is also the main purpose of our research.

Point 3: On the other hand, when providing the results of study 1, the authors should emphasize packaging design more since it is the factor treated in another two studies.

Response 3: Thank you for the suggestion. In response to your suggestion, we have included discussions related to packaging design in some of the texts. Here is what we added.

Actual version

Using the semantics of the vocabulary within the topics, the probability value of the vocabulary, and the literature data, we identified three topics: quality of service, quality of food, and perceived value. The report indicates that when these three factors do not meet consumers' expectations when ordering and eating takeaway food, there is a high likelihood of negative feedback. This is also the basis of both study 2 and study 3, our research mainly focuses on whether adding augmented reality technology to packaging design will improve these negative comments.

Meanwhile, we discovered that the meaning of the title had been accidentally mistranslated during the translation of the previous manuscript. Our research, in fact, focuses more on the effect of packaging on improving all possible negative consumer evaluations in general. It is possible that you misunderstood that we primarily focus on negative reviews due to packaging issues. Please accept our sincere apologies for this translation error. The title has been adjusted in this revision.

Previous version

A study of how augmented reality improves young consumers' negative evaluations of takeaway food packaging

Actual version

Effects of using augmented reality technology in takeaway food packaging to improve young consumers' negative evaluations

Point 4: Finally, since implementing moderation analysis, the authors should formulate appropriate hypotheses and provide literature background for them as well.

Response 4: Thank you for the suggestion. The research hypotheses were added in accordance with your suggestion. In addition, a section explaining why we want to test the moderation effect is added to the literature review and research hypotheses chapter. Below is a revised version of the text.

Actual version

In this study, a quantitative model is developed to assess the effect of augmented reality packaging on improving negative evaluations. The study examines the perception and behavior of consumers when augmented reality packaging is used. The study examines the technology incentive model, satisfaction, and purchase intention. A relational framework is proposed to describe path relationships among constructs, and all hypotheses are tested. Figure 2 shows the hypothesized model. According to the extended unified theory of acceptance and use of technology (UTAUT2), gender, age, and experience of consumers may be important moderating factors[22]. There has been a huge impact of this theoretical model in the field of user research[23]. Thus, we examined the moderating effects of these three variables for each path relationship in our study.

Point 5: The sample – the authors should add (in the part of the text in which they present characteristics of the sample of respondents) the type of the sample and sampling frame.

Response 5: Thank you for the suggestion. As you suggested, we have added a description of the expected respondents to the sampling results before presenting them.

Actual version

We conducted our survey in China. We mainly target young consumers since they order takeaway food more frequently and are more likely to be familiar with technology products. As can be seen from the sampling results, marital status, income, education level, and occupation type are more representative of the general basic situation of Chinese youth.The basic information of the subjects is presented in Table 1. In study 2, the subjects are recruited via the Internet to take part in the survey, with two measurements being repeated on each subject.

Point 6: Authors need to specify when selecting “ 3 national chain catering brands that currently offer food delivery services“ on the basis of “ a large number of negative evaluations on the food delivery platform” whether those negative evaluations were in regard to their packaging design or generally.

Response 6: Thank you for the suggestion. It is important to note that these negative reviews are for the general case. We appreciate your academic rigor in separating negative perceptions from packaging from negative perceptions in general. Throughout this article, we hope that our findings will have wider implications for management, so we do not intend solely to address negative customer evalutaions resulting from poor packaging design. The purpose of this study is to explore and verify whether negative user evaluation results in general can be influenced by packaging design, and especially packaging design that incorporates augmented reality technology. We have modified the text to clarify this paragraph with your assistance.

Actual version

Based on the large number of negative evaluations on the food delivery platform, we selected 3 national chain catering brands that currently offer food delivery services. In general, these selected brands were receiving more negative evaluations than their peers, which is not just due to poor packaging design. In order to incorporate augmented reality technology into some of these brands' product packages, we designed augmented reality packages based on some of their product packages.

Point 7: Since during conducting study 2 there were “two measurements being repeated on each subject” whereby “the subjects completed the questionnaire for the original packaging, then observed the augmented reality packaging and completed the second questionnaire” it remains unclear how they provided answers for the three constructs considering original packaging when all the items for these constructs provided in appendix A are formulated from the context of implementing AR.

Response 7: Thank you for the suggestion. We are glad about this suggestion and modified the items as follow:

TR1: I can count on this company/companies that use AR takeaway packaging to consider how their actions will affect customers like me. (Deleted)

TR2: If I were to have any problems, this company/companies that use AR takeaway packaging will be ready and willing to offer me assistance and support.

TR3: When making decisions about its policies, this company/companies that use AR takeaway packaging is concerned about customers like me.

TR4: I can count on this company/companies that use AR takeaway packaging to be sincere in its communication.

TR5: Even if this company/companies that use AR takeaway packaging were to provide an unlikely explanation, I would be confident that the explanation was correct. (Deleted)

SA1: I am satisfied with takeaways of this company/companies that use AR takeaway packaging.

SA2: Considering all my experience with food, my choice of takeaway of this company/companies that use AR takeaway packaging was wise.

SA3: Overall, I am pleased with takeaway of this company/companies that use AR takeaway packaging based on my experience.

PuI1: I regard of this company/companies that use AR takeaway packaging as my first choice for takeaway purchases.

PuI2: I plan to order more takeaways from of this company/companies that use AR takeaway packaging in the next few years.

PuI3: I will continue to buy takeaways at of this company/companies that use AR takeaway packaging in the next few years.

Point 8: The relationships between first-order constructs (PI, PV, NE) and the second-order construct (Technology incentives) – the authors should consider whether that relation is reflective or formative. In the case of the second approach, which seems more appropriate to me, the data analysis would need to be performed once again relying on, for example, SmartPLS software. When providing such discussions the authors can rely, among others, on: Jarvis, C. B., MacKenzie, S. B., Podsakoff, P. M. (2003). A Critical Review of Construct Indicators and Measurement Model Misspecification in Marketing and Consumer Research. Journal of Consumer Research, 30 (2), 199-218.

Response 8: Thank you for the suggestion. Your recommendations regarding the literature were carefully reviewed and we are grateful for your assistance. It is important to take into consideration this relationship when using the second-order construct, as we agree with your standpoint. We therefore followed your advice and used Smart PLS to calculate the model. After calculated and deliberate consideration, we did not decide to replace the reflective relationship with a formative relationship. There are three main reasons.

First, the calculation indicated that the results of the modified model failed to fit well. The calculation of the path relationship in PLS under the premise of using the reflective relationship obtained similar results to those obtained in AMOS. It should be noted, however, that when the arrow is reversed, the factor loading of the first-order constructs to the second-order constructs has dropped significantly.

Second, the second-order confirmatory factor analysis has proven stable and has been backed up by previous research. Based on the reflective relationship, we conducted a second-order confirmatory factor analysis on TI alone, and obtained good model fit results. Moreover, the reflective relationship has also been used in previous research regarding the establishment of the technology incentive model we referred to.

Lastly, we rely on the literature judgment criteria that you provided. According to the article: More specifically, a construct should be modeled as having formative indicators if the following conditions prevail: (a) the indicators are viewed as defining characteristics of the construct, (b) changes in the indicators are expected to cause changes in the construct, (c) changes in the construct are not expected to cause changes in the indicators, (d) the indicators do not necessarily share a common theme, (e) eliminating an indicator may alter the conceptual domain of the construct, (f) a change in the value of one of the indicators is not necessarily expected to be associated with a change in all of the other indicators, and (g) the indicators are not expected to have the same antecedents and consequences. On the other hand, a construct should be modeled as having reflective indicators if the opposite is true and the conditions shown in the last column in the table are satisfied.

As a result, we believe that most of these criteria are unclear for the second-order constructs in this study, and it is difficult to demonstrate that the constructs do not follow the reflective relationship. In addition, it should be noted that (d) and (f) are relatively easy to evaluate. Across both criteria, our study found significant correlations between the constructs, which may indicate that our model does not apply to conditions classified as formative.

However, we are grateful for your suggestion. In the course of verifying the way the model was constructed, we also rechecked the validity of the samples. In this round of revisions, we removed 32 invalid samples that should have been deleted and recalculated all the data. The current version has been thoroughly screened and recalculated, resulting in a more sensible and rigorous analysis of the data.

Point 9: Naming of the constructs in the context of the items from the questionnaire – The authors adapted the questionnaire to AR packaging design. For example, one of the items measuring food quality is formulated in the following manner: “QoF1: I think the AR takeaway packaging helps the food to have a good presentation.” The authors should justify whether it is appropriate to name that construct as Food quality since it actually measures Food quality from the aspect of AR takeaway packaging (or perceived influence of AR takeaway packaging on food quality), and not in general. There can be found an analogy in the case of other constructs as well.

Response 9: Thank you for the suggestion. As indicated in the reference, this item is derived from the method of measuring the quality of food. In our reference literature, four items are used to measure the quality of food, including presentation, variety, taste, and health. In this study, the presentation was adapted to reflect the QoF1. This question is intended to investigate the perceptions of users regarding food presentation. We have also reviewed the questionnaire thoroughly in response to your suggestions. We are interested in the measurement dimensions that each construct possesses. This premise indicates that "a company or product that uses AR technology to make packaging" is considered an attribute. It is used in order to emphasize who is being asked for information as well as to determine consumer perceptions of what is being asked for.

Point 10: In addition to all previous comments, I could add some technical issues (for reference 21, there is not provided the name of the scientific journal; generally, links with doi numbers are missing; in row 199, there is a sign I believe from a Chinese alphabet).

Response 10: Thank you for the suggestion. Your reminder is greatly appreciated. We have revised the journal name for all articles and added the DOI. Additionally, we removed 199 lines of Chinese characters.

Reviewer 2 Report

The subject of the article is interesting but some clarifications should be made to the article’s content.

1.      The research questions or objectives are not very clear stated. The necessity of the research should result from these objectives.

2.      The Research methodology section should be improved in order to be very clear what Research methods were used. For the first study the research method was not mentioned. The software used and the process of data analysis should be explained, in order to be very clear for readers how the researchers conducted the study.

3.      The role of AR is not very clear. For what purpose it is used? How does it help consumers in their purchasing decision? Is the price of the product affected? Were the consumers informed of a possible higher price?

4.      In the Results section, for Study 2 it is not mentioned the method used for data analysis. Maybe this method should be included in the Methodology and mentioned when the results are presented. Since this research seems to be an experiment, it is not very clear how this one was conducted.

5.      The Conclusion and future work section should state very clear what are the study implications for theory development and other actors. There are some statements about the utility of AR for business but what implications could be in terms of costs, investments or package recycling. The implication for theory should also be in-depth described.

As a general conclusion, the authors have to make some revisions in order to improve the quality of this article.

Author Response

Response to Reviewer 2 Comments

Dear Referee,

Thank you very much for the valuable suggestions, which have made our paper more comprehensive. We have responded to each of your comments below:

Point 1: The research questions or objectives are not very clear stated. The necessity of the research should result from these objectives.

Response 1: Thank you for the suggestion. In response to your suggestion, we added the necessity and purpose of the study in the section on research purposes. Below are the additions we made.

Actual version

To assist restaurants in improving the performance of food delivery services in retail, it is imperative to improve the negative ratings of consumers in the delivery region. As augmented reality technology has been proven beneficial to marketing in numerous industries and activities, this study examines whether application of this technology in takeaway food packaging can improve consumer negative evaluations, as well as how consumer behavior is affected. Researchers conducted 3 studies to evaluate whether the application of augmented reality technology to take-out packaging would help to improve consumers' negative perceptions of these products. Figure 1 illustrates the research process. As part of Study 1, we conducted topic modeling using Latent Dirichlet Allocation to summarize the main causes of negative evaluations. We conducted Study 2 to explore whether augmented reality packaging could prove helpful for consumers by selecting a random product packaging sample and creating augmented reality samples to compare the perceived difference between ordinary packaging and augmented reality packaging. In Study 3, we developed a structural equation model based on a technology incentive model in order to theoretically verify the effect of augmented reality packaging on negative evaluations.

Point 2: The Research methodology section should be improved in order to be very clear what Research methods were used. For the first study the research method was not mentioned. The software used and the process of data analysis should be explained, in order to be very clear for readers how the researchers conducted the study.

Response 2: Thank you for the suggestion. Following your comments, we have included a discussion of calculation methods in Chapter 3. Here is what we added.

Actual version

The text data is analyzed using Python, with the gensim lda module estimating Latent Dirichlet Allocation models using our corpus of text data. ince the number of topics must be set manually, we introduce two parameters of perplexity and coherence to determine the optimal number of topics for the topic model. A smaller perplexity value indicates a better result, according to the derivation formula (1) [93]. In addition, c value calculates the score using normalized pointwise mutual information (NPMI) and cosine similarity between words in the content vector. It can be concluded that the higher the value, the better [94].

Point 3: The role of AR is not very clear. For what purpose it is used? How does it help consumers in their purchasing decision? Is the price of the product affected? Were the consumers informed of a possible higher price?

Response 3: Thank you for the suggestion. Based on your feedback, chapter 3 has been updated to include augmented reality. We conducted this research to investigate and verify whether augmented reality technology could have an impact on consumer behavior. The modified results are presented below.

Actual version

In study 2 and study 3, Unity Vuforia was used to create augmented reality packaging. Currently, this is one of the most popular engines for designing augmented reality applications [95]. Augmented reality technology can be used to promote brands, introduce ingredients and cooking methods, introduce promotional information, and provide consumers with an interactive experience.There is a possibility that the development of augmented reality packaging will require higher costs. These costs will depend on factors such as prices and level of development in different countries, which requires further research. Upon equal distribution of the production costs, the unit price may not be significantly impacted by the production costs if there are a large number of sales. Hence, the purpose of this study is not to estimate the unit price of augmented reality packaging versus traditional packaging, but merely to assess the impact of this technology on consumer behavior.

Point 4: In the Results section, for Study 2 it is not mentioned the method used for data analysis. Maybe this method should be included in the Methodology and mentioned when the results are presented. Since this research seems to be an experiment, it is not very clear how this one was conducted.

Response 4: Thank you for the suggestion. Thanks for reminding us that this was indeed a discursive omission that occurred during revisions. We have supplemented the research methods section with relevant discussion.

Actual version

The basic information of the subjects is presented in Table 1. In study 2, the subjects are recruited via the Internet to take part in the survey, with two measurements being repeated on each subject. In order to compare consumer trust, satisfaction, and purchase intentions for takeaway foods using original packaging and augmented reality packaging, we used multivariate analysis of variance (MANOVA). Initially, the subjects completed the questionnaire for the original packaging, then observed the augmented reality packaging and completed the second questionnaire. A total of 165 questionnaires were collected during the period of January to March 2022. As a check on the logic of the respondents, we set reverse questions, and judged their concentration based on the time they spent answering the questions. After excluding invalid samples, the remaining 104 valid questionnaires were used for data analysis. The effective rate of the questionnaire was 63.030%.

In addition, we have supplemented the research results chapter with the following information.

Actual version

This stage of research examines whether the use of augmented reality technology in take-out packaging will influence consumer preferences. In other words, to determine whether there is a difference in consumer evaluation results when augmented reality packaging is compared to traditional packaging. MANOVA is used to determine if there are significant differences between consumers' perceptions of trust, satisfaction, and purchase intention for two types of packaged takeaways, in order to test the effectiveness of augmented reality technology in improving consumer negative evaluations.

Point 5: The Conclusion and future work section should state very clear what are the study implications for theory development and other actors. There are some statements about the utility of AR for business but what implications could be in terms of costs, investments or package recycling. The implication for theory should also be in-depth described.

Response 5: Thank you for the suggestion. The management and theoretical contributions have been supplemented in more detail in the conclusion section following your suggestion. The following are the details.

Actual version

Specifically, restaurants with a large number of negative consumer evaluations should consider paying close attention to packaging design, especially using augmented reality technology to address their issues. The results of our study demonstrate that interactive packaging can improve consumers' negative perceptions of takeaway food and improve their evaluations of it. Moreover, this improvement encompasses a variety of aspects, not only addressing insufficient packaging design as a source of negative evaluations. While the results of the comparison of building models based on gender, age, and whether they have interactive experience using augmented reality show that classification characteristics of consumers do not cause changes in consumer behavior in most cases, women and consumers with more interactive experience still responded positively to augmented reality marketing strategies. Thus, restaurants that offer takeaway services should first analyze their own consumer characteristics. It will be more likely to consider the actual management strategy of using AR as a package to minimize negative consumer feedback in the event that a significant portion of the main consumer group matches the characteristics of women and AR interactive experience.

Our theoretical contributions include the proposed reasons for consumers' negative reviews based on the results of text analysis performed using unsupervised machine learning methods, along with the development of a theoretical framework based on the technology incentive model. A verified version of our new theoretical model is shown in Figure 9, and we refer to it as the expanded technology incentive model (ETIM).The theoretical model we propose may be applicable to human-computer interactions in broader fields of food marketing, and is not limited to augmented reality applications. The framework examines three main characteristics of interactive products, namely perceived interactivity, perceived vividness, and novelty experience, and evaluates the impact of this experience on consumer behavior [16].As shown by our verification results, augmented reality technology is an effective strategy for restaurants that offer takeaway food. The perception of interactivity, perceived vividness, and novelty experience all contribute to marketing success. While it has been established that there is no direct and significant relationship between consumers' flow experience and trust, the incentive experience of human-computer interaction has been shown to improve consumers' antecedents of evaluation in order to ultimately increase the effect of purchase intention.

Moreover, we discuss the impact of augmented reality on cost, investment, or package return based on your suggestions.

Actual version

In addition to solving the three problems mentioned above, the comparison of cost increases and return benefits associated with augmented reality packaging is also one of the scopes that can be examined for future research. It is inevitable that the adoption of augmented reality technology will increase the design costs during the process of in-depth development and iteration [139].It is worth evaluating whether restaurants can reduce unit price costs through a large number of sales. Fortunately, unlike other interactive devices, augmented reality technology supports mobile phones and tablets, which also helps to reduce the hardware cost [140].Catering companies that have been developing augmented reality packaging for a considerable period of time may be able to lower the cost of a single augmented reality food packaging to a level similar to that of traditional packaging. It is, however, necessary to conduct further evaluations. In addition, packaging sustainability is an important issue that should be taken into consideration[141].With augmented reality technology, there is no additional hardware requirement, which makes it a more environmentally friendly and sustainable alternative to other complex interactive experiences. The user must, however, actively download the software launched by the restaurant before engaging in any interaction. It is possible for some users to express distrust toward third-party software[142]. Furthermore, users may be unwilling to download additional software as they are concerned about their mobile device's capacity [143]. In addition, this indicates that guiding users to download software launched by catering brands, and maintaining their willingness to continue using these products, are important marketing issues that should be investigated further.

Reviewer 3 Report

The title “A study of how augmented reality improves young consumers' negative evaluations of takeaway food packaging” is interesting and according to special issue of journal.

If you concentrate on the following points, your paper will look more delectable.

1.     Please enhance the Abstract section with results and words.

2.     Please add more keywords like marketing, consumers, etc.

3.     Introduction and literature sections looks good.

Please cite these 2 articles

https://doi.org/10.3390/su13126839

https://doi.org/10.3390/su131910705

4.     Method is also looking good.

5.     Please add the more paragraph in Conclusion section.

6.     Please add the recommendations section.

7.     Please add the future implications section as well.

8.     Please recheck the citations and references.

All the best for the next step. 

Author Response

Response to Reviewer 3 Comments

Dear Referee,

Thank you very much for the valuable suggestions, which have made our paper more comprehensive. We have responded to each of your comments below:

Point 1: Please enhance the Abstract section with results and words.

Response 1: Thank you for the suggestion. Our Abstract has been revised in order to present the findings more clearly and specifically.

Actual version

This paper examines the use of augmented reality technology in the design of packaging for takeaway food to assist in marketing. The research is divided into 3 studies for progressive investigation and analysis. Study 1 collected 375,859 negative evaluations of food delivery from the Internet and explored the main reasons that may impact the user's evaluation by Latent Dirichlet Allocation topic modeling. Study 2 evaluated the effectiveness of augmented reality packaging by surveying 165 subjects and comparing it with traditional packaging. We conducted a survey of 1603 subjects in study 3 and used the technology incentive model (TIM) to analyze how augmented reality technology positively impacts food delivery marketing. It has been established that packaging will influence the negative perception of consumers about buying and eating takeout food. Specifically, augmented reality technology can improve negative evaluations, by providing a more conducive user experience than traditional packaging. According to our findings, augmented reality technology has improved the consumers' perception of interaction, perceived vividness, and novelty experience, and achieved the aim of promoting takeaway food retail by improving negative evaluations posted by users.

Point 2: Please add more keywords like marketing, consumers, etc.

Response 2: Thank you for the suggestion. Following your suggestion, we have added keywords.

Point 3: Introduction and literature sections looks good. Please cite these 2 articles.

Response 3: Thank you for the suggestion. We appreciate your confirmation that both articles have been cited.

Point 4: Method is also looking good.

Response 4: Thank you for the suggestion. Please accept my sincere thanks once again for your kind words.

Point 5: Please add the more paragraph in Conclusion section.

Response 5: Thank you for the suggestion. In the Conclusions section, we have added a paragraph describing theoretical contributions in response to your suggestion. Below are the additions we made.

Actual version

Our theoretical contributions include the proposed reasons for consumers' negative reviews based on the results of text analysis performed using unsupervised machine learning methods, along with the development of a theoretical framework based on the technology incentive model. A verified version of our new theoretical model is shown in Figure 9, and we refer to it as the expanded technology incentive model (ETIM). The theoretical model we propose may be applicable to human-computer interactions in broader fields of food marketing, and is not limited to augmented reality applications. The framework examines three main characteristics of interactive products, namely perceived interactivity, perceived vividness, and novelty experience, and evaluates the impact of this experience on consumer behavior [16].As shown by our verification results, augmented reality technology is an effective strategy for restaurants that offer takeaway food. The perception of interactivity, perceived vividness, and novelty experience all contribute to marketing success. While it has been established that there is no direct and significant relationship between consumers' flow experience and trust, the incentive experience of human-computer interaction has been shown to improve consumers' antecedents of evaluation in order to ultimately increase the effect of purchase intention.

Point 6: Please add the recommendations section.

Response 6: Thank you for the suggestion. According to your suggestion, we have added a new paragraph in the conclusion section describing the practical management recommendations. Here is what we added.

Actual version

Specifically, restaurants with a large number of negative consumer evaluations should consider paying close attention to packaging design, especially using augmented reality technology to address their issues. The results of our study demonstrate that interactive packaging can improve consumers' negative perceptions of takeaway food and improve their evaluations of it. Moreover, this improvement encompasses a variety of aspects, not only addressing insufficient packaging design as a source of negative evaluations. While the results of the comparison of building models based on gender, age, and whether they have interactive experience using augmented reality show that classification characteristics of consumers do not cause changes in consumer behavior in most cases, women and consumers with more interactive experience still responded positively to augmented reality marketing strategies. Thus, restaurants that offer takeaway services should first analyze their own consumer characteristics. It will be more likely to consider the actual management strategy of using AR as a package to minimize negative consumer feedback in the event that a significant portion of the main consumer group matches the characteristics of women and AR interactive experience.

Point 7: Please add the future implications section as well.

Response 7: Thank you for the suggestion. Following your suggestion, we have added a section on future impacts to the conclusion. Here is what we added.

Actual version

In addition to solving the three problems mentioned above, the comparison of cost increases and return benefits associated with augmented reality packaging is also one of the scopes that can be examined for future research. It is inevitable that the adoption of augmented reality technology will increase the design costs during the process of in-depth development and iteration [139].It is worth evaluating whether restaurants can reduce unit price costs through a large number of sales. Fortunately, unlike other interactive devices, augmented reality technology supports mobile phones and tablets, which also helps to reduce the hardware cost [140].Catering companies that have been developing augmented reality packaging for a considerable period of time may be able to lower the cost of a single augmented reality food packaging to a level similar to that of traditional packaging. It is, however, necessary to conduct further evaluations. In addition, packaging sustainability is an important issue that should be taken into consideration[141].With augmented reality technology, there is no additional hardware requirement, which makes it a more environmentally friendly and sustainable alternative to other complex interactive experiences. The user must, however, actively download the software launched by the restaurant before engaging in any interaction. It is possible for some users to express distrust toward third-party software[142]. Furthermore, users may be unwilling to download additional software as they are concerned about their mobile device's capacity [143]. In addition, this indicates that guiding users to download software launched by catering brands, and maintaining their willingness to continue using these products, are important marketing issues that should be investigated further.

Point 8: Please recheck the citations and references.

Response 8: Thank you for the suggestion. References have been reviewed and corrected.

Round 2

Reviewer 2 Report

The authors performed the recommended improvements.